# Behavior Alignment via Reward Function Optimization

**Dhawal Gupta**[*]
University of Massachusetts

**Yash Chandak**[*][†]
Stanford University

**Scott M. Jordan**[†]
University of Alberta

**Philip S. Thomas**
University of Massachusetts

**Bruno Castro da Silva**
University of Massachusetts

## Abstract

Designing reward functions for efficiently guiding reinforcement learning (RL) agents toward specific behaviors is a complex task. This is challenging since it requires the identification of reward structures that are not sparse and that avoid inadvertently inducing undesirable behaviors. Naively modifying the reward structure to offer denser and more frequent feedback can lead to unintended outcomes and promote behaviors that are not aligned with the designer's intended goal. Although potential-based reward shaping is often suggested as a remedy, we systematically investigate settings where deploying it often significantly impairs performance. To address these issues, we introduce a new framework that uses a bi-level objective to learn *behavior alignment reward functions*. These functions integrate auxiliary rewards reflecting a designer's heuristics and domain knowledge with the environment's primary rewards. Our approach automatically determines the most effective way to blend these types of feedback, thereby enhancing robustness against heuristic reward misspecification. Remarkably, it can also adapt an agent's policy optimization process to mitigate suboptimalities resulting from limitations and biases inherent in the underlying RL algorithms. We evaluate our method's efficacy on a diverse set of tasks, from small-scale experiments to high-dimensional control challenges. We investigate heuristic auxiliary rewards of varying quality— some of which are beneficial and others detrimental to the learning process. Our results show that our framework offers a robust and principled way to integrate designer-specified heuristics. It not only addresses key shortcomings of existing approaches but also consistently leads to high-performing solutions, even when given misaligned or poorly-specified auxiliary reward functions.

## 1 Introduction

In this paper, we investigate the challenge of enabling reinforcement learning (RL) practitioners, who may not be experts in the field, to incorporate domain knowledge through heuristic auxiliary reward functions. Our goal is to ensure that such auxiliary rewards not only induce behaviors that align with the designer's intentions but also allow for faster learning. RL practitioners typically model a given control problem by first designing simple reward functions that directly quantify whether (or how well) an agent completed a task. These could be, for instance, functions assigning a reward of $+1$ iff the agent reaches a specified goal state, and zero otherwise. However, optimizing a policy based on such a sparse reward function often proves challenging.

---

[*]Both authors contributed equally to this work. [†] Work done while at the University of Massachusetts. Corresponding author: Dhawal Gupta (`dgupta@cs.umass.edu`).

37th Conference on Neural Information Processing Systems (NeurIPS 2023).

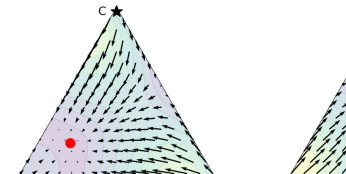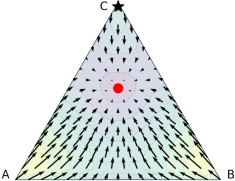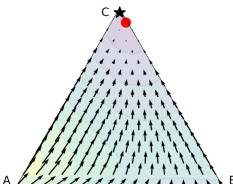

Figure 1: Auxiliary rewards can be used to convey to the agent how we (designers) *think* it should solve the problem. However, if not carefully designed, they can lead to policies that result in undesired behaviors. This figure provides a visual illustration of a toy example depicting how the proposed method works. The star represents the optimal policy, and the red dot represents the fixed point of a policy optimization process under a "sub-optimal" heuristic; i.e., one that, when naively combined with $r_p$, induces behaviors different from those under the optimal policy for $r_p$. **(Left)** Vector field of a policy optimization process converging to a sub-optimal policy. **(Middle and Right)** By changing the influence of auxiliary rewards, our method can dynamically *correct* the *entire policy optimization process* steering it towards a policy that results in the desired behavior.

To address this issue, designers often introduce auxiliary reward functions that supplement the original rewards. Auxiliary rewards are heuristic guidelines aimed at facilitating and speeding up the learning process. One could, e.g., augment the previously described reward function (which gives a reward of $+1$ upon reaching a goal state) with an auxiliary reward accounting for the agent's distance to the goal. However, the effectiveness of using auxiliary reward functions largely depends on the problem's complexity and the designer's skill in crafting heuristics that, when combined with the original reward function, do not induce behaviors different than the ones originally intended [26, 27].

Existing methods like potential-based reward shaping [41] aim to incorporate domain knowledge without misaligning the behaviors induced by the resulting combined reward functions. However, as we discuss in Section 3, potential-based shaping has several limitations: *(i)* it is restricted to state-based functions; *(ii)* it amounts to a different initialization of the $q$-function; *(iii)* it does not alter policy gradients in expectation; and *(iv)* it can increase the variance in policy gradient methods.

To address these challenges, we introduce a scalable algorithm that empowers RL practitioners to specify potentially imperfect auxiliary reward functions. It ensures that the resulting optimization process will not inadvertently lead to unintended behaviors and that it will allow for faster learning. In particular, this paper addresses the following challenges:

**(1) How to incorporate auxiliary reward information:** We introduce a novel bi-level objective to analyze and automatically fine-tune designer-created auxiliary reward functions. It ensures they remain aligned with the original reward and do not induce behaviors different from those originally intended by the designer. Additionally, we formulate the problem to shape the optimization landscape, biasing our bi-level optimizer toward auxiliary reward functions that facilitate faster learning.

**(2) How to use auxiliary reward to mitigate algorithmic biases:** We show that our framework can automatically adjust how primary and auxiliary rewards are blended to mitigate limitations or biases inherent in the underlying RL algorithm (Section 4.1). For instance, many policy-gradient-based RL algorithms are subject to biases due to issues like discounting mismatch [59] or partial off-policy correction [51]. These biases can hinder the algorithm's ability to identify near-optimal policies.

**(3) How to ensure scalability to high-dimensional problems:** We introduce an algorithm that employs *implicit gradients* to automatically adjust primary and auxiliary rewards, ensuring that the combined reward function aligns with the designer's original expectations (see Figure 1). We evaluate our method's efficacy across a range of tasks, from small-scale to high-dimensional control settings (see Section 6). In these tasks, we experiment with auxiliary rewards of varying quality; some accelerate learning, while others can be detrimental to finding an optimal policy.

## 2 Notation

In this paper, we investigate sequential decision-making problems modeled as Markov decision processes (MDPs). An MDP is defined as a tuple $(\mathcal{S}, \mathcal{A}, p, r_p, r_{\text{aux}}, \gamma, d_0)$, where $\mathcal{S}$ is the state set, $\mathcal{A}$ is the action set, $p$ is the transition function, $r_p : \mathcal{S} \times \mathcal{A} \to \mathbb{R}$ is the *primary* reward function, $r_{\text{aux}} :$

$S \times A \to \mathbb{R}$ is an *optional* auxiliary reward function (possibly designed by a non-expert in machine learning, based on domain knowledge), and $d_0$ is the starting state distribution. Let $\pi_\theta : S \times A \to [0, 1]$ be any policy parameterized using $\theta \in \Theta$. For brevity, we will often use $\pi_\theta$ and $\theta$ interchangeably. Let $S_t$ and $A_t$ be the random variables for the state and action observed at the time $t$. As in the standard RL setting, the performance $J(\theta)$ of a policy $\pi_\theta$ is defined as the expected discounted return with respect to the (primary) reward function, $r_p$; i.e., $J(\theta) := \mathbb{E}_\pi[\sum_{t=0}^{T} \gamma^t r_p(S_t, A_t)]$, where $T + 1$ is the episode length. An optimal policy parameter $\theta^*$ is defined as $\theta^* \in \arg\max_{\theta \in \Theta} J(\theta)$. A popular technique to search for $\theta^*$ is based on constructing sample estimates $\hat{\Delta}(\theta, r_p)$ of the ($\gamma$-dropped) policy gradient, $\Delta(\theta, r_p)$, given an agent's interactions with the environment for one episode [58, 59]. Then, using $\psi_\theta(s, a)$ as a shorthand for $\mathrm{d} \ln \pi_\theta(s, a)/\mathrm{d}\theta$, these quantities are defined as follows:

$$\Delta(\theta, r_p) = \mathbb{E}_{\pi_\theta}\left[\hat{\Delta}(\theta, r_p)\right] \quad \text{and} \quad \hat{\Delta}(\theta, r_p) := \sum_{t=0}^{T} \psi_\theta(S_t, A_t) \sum_{j=t}^{T} \gamma^{j-t} r_p(S_j, A_j). \quad (1)$$

## 3 Limitations of Potential Based Reward Shaping

When the objective function $J(\theta)$ is defined with respect to a *sparse* reward function $r_p$ (i.e., a reward function such that $r_p(s, a) := 0$ for most $s \in S$ and $a \in A$), searching for $\theta^*$ is challenging [24]. A natural way to provide more frequent (i.e., denser) feedback to the agent, in the hope of facilitating learning, is to consider an alternate reward function, $\tilde{r}_{\text{naive}} := r_p + r_{\text{aux}}$. However, as discussed earlier, $r_{\text{aux}}$ may be a designer-specified auxiliary reward function not perfectly aligned with the objective encoded in $r_p$. In this case, using $\tilde{r}_{\text{naive}}$ may encourage undesired behavior. An alternative way to incorporate domain knowledge to facilitate learning was introduced by Ng et al. [41]. They proposed using a *potential function*, $\Phi : S \to \mathbb{R}$ (analogous to $r_{\text{aux}}$), to define new reward functions of the form $\tilde{r}_\Phi(S_t, A_t, S_{t+1}) := r_p(S_t, A_t) + \gamma\Phi(S_{t+1}) - \Phi(S_t)$. Importantly, they showed that optimal policies with respect to the objective $\mathbb{E}[\sum_{t=0}^{T} \gamma^t \tilde{r}_\Phi(S_t, A_t, S_{t+1})]$ are also optimal with respect to $J(\theta)$.

While potential-based reward shaping can partially alleviate some of the difficulties arising from sparse rewards, Wiewiora [64] showed that $q$-learning using $\tilde{r}_\Phi$ produces the *exact same sequence of updates* as $q$-learning using $r_p$ but with a different initialization of $q$-values. In what follows, we establish a similar result: we show that performing potential-based reward shaping has *no impact on expected policy gradient* updates; and that it can, in fact, even increase the variance of the updates.

**Property 1.** $\mathbb{E}[\hat{\Delta}(\theta, \tilde{r}_\Phi)] = \mathbb{E}[\hat{\Delta}(\theta, r_p)]$ *and* $\mathrm{Var}(\hat{\Delta}(\theta, \tilde{r}_\Phi))$ *can be higher than* $\mathrm{Var}(\hat{\Delta}(\theta, r_p))$.

All proofs are deferred to Appendix A. The above points highlight some of the limitations of potential-based shaping for policy gradients and $q$-learning—both of which form the backbone of the majority of model-free RL algorithms [58]. Furthermore, potential functions $\Phi$ *cannot* depend on actions [41], which restricts the class of eligible auxiliary rewards $r_{\text{aux}}$ and heuristics functions that may be used. Finally, notice that $\Phi$ is designed independently of the agent's underlying learning algorithm. As we will show in the next sections, our method can autonomously discover auxiliary reward functions that not only facilitate learning but also help mitigate various types of algorithmic limitations and biases.

## 4 Behavior Alignment Reward Function

In this section, we introduce an objective function designed to tackle the primary challenge investigated in this paper: how to effectively leverage designer-specified auxiliary reward functions to rapidly induce behaviors envisioned by the designer. The key observation is that naively adding an auxiliary reward function $r_{\text{aux}}$ to $r_p$ may produce policies whose corresponding behaviors are misaligned with respect to the behaviors induced by $r_p$. In these cases, $r_{\text{aux}}$ should be ignored during the search for an optimal policy. On the other hand, if $r_p$ and $r_{\text{aux}}$ may be combined in a way that results in the desired behaviors, then combinations that produce frequent and informative feedback to the agent should be favored, as they are likely to facilitate faster learning.

To tackle the challenges discussed above, we employ a bi-level optimization procedure. This approach aims to create a *behavior alignment reward* by combining $r_{\text{aux}}$ and $r_p$ using a parameterized function. Our method is inspired by the optimal rewards framework by Singh et al. [52, 53]. Let $\gamma_\varphi \in [0, 1)$ be

a *discount rate value* parameterized by $\varphi \in \Gamma$.[2] Let $r_\phi : \mathcal{S} \times \mathcal{A} \to \mathbb{R}$ be a *behavior alignment reward*: a function of both $r_p$ and $r_{\text{aux}}$, parameterized by $\phi \in \Upsilon$, where $\Upsilon$ and $\Gamma$ are function classes. One example of a behavior alignment reward function is $r_\phi(s,a) \coloneqq f_{\phi_1}(s,a) + \phi_2 r_p(s,a) + \phi_3 r_{\text{aux}}(s,a)$, where $f_\phi : \mathcal{S} \times \mathcal{A} \to \mathbb{R}$ and $\phi \coloneqq (\phi_1, \phi_2, \phi_3)$. Let `Alg` be any gradient/semi-gradient/non-gradient-based algorithm that outputs policy parameters. To mitigate possible divergence issues arising from certain policy optimization algorithms like DQN [60, 1], we make the following simplifying assumption, which can generally be met with appropriate regularizers and step sizes:

**Assumption 1.** *Given $r_\phi$ and $\gamma_\varphi$, the algorithm* `Alg`$(r_\phi, \gamma_\varphi)$ *converges to a fixed point $\theta$— which we denote as $\theta(\phi, \varphi) \in \Theta$ to emphasize its indirect dependence on $\phi$ and $\varphi$ through* `Alg`, *$r_\phi$, and $\gamma_\varphi$.*

Given this assumption, we now specify the following bi-level objective:

$$\phi^*, \varphi^* \in \underset{\phi \in \Upsilon, \varphi \in \Gamma}{\arg\max} \quad J(\theta(\phi, \varphi)) - \lambda_\gamma \gamma_\varphi, \quad \text{where} \quad \theta(\phi, \varphi) \coloneqq \texttt{Alg}(r_\phi, \gamma_\varphi). \tag{2}$$

Here, $\lambda_\gamma$ serves as the regularization coefficient for the value of $\gamma_\varphi$, and `Alg` denotes a given policy optimization algorithm. Let, as an example, `Alg` be an on-policy gradient algorithm that uses samples to estimate the gradient $\Delta(\theta, r_p)$, as in (1). We can then define a corresponding variant of $\Delta(\theta, r_p)$ that is compatible with our formulation and objective, and which uses both $r_\phi$ and $\gamma_\varphi$, as follows:

$$\Delta_{\text{on}}(\theta, \phi, \varphi) \coloneqq \mathbb{E}_{\pi_\theta} \left[ \sum_{t=0}^{T} \psi_\theta(S_t, A_t) \sum_{j=t}^{T} \gamma_\varphi^{j-t} r_\phi(S_j, A_j) \right]. \tag{3}$$

Notice that the bi-level formulation in (2) is composed of three key components: outer and inner objectives, and an outer regularization term. In what follows, we discuss the need for these.

**Need for Outer- and Inner-Level Objectives:** The **outer-level objective** in Equation (2) serves a critical role: it evaluates different parameterizations, denoted by $\phi$, for the behavior alignment reward function. These parameterizations influence the induced policy $\theta(\phi, \varphi)$, which is evaluated using the performance metric $J$. Recall that this metric quantifies the alignment of a policy with the designer's primary reward function, $r_p$. In essence, the outer-level objective seeks to optimize the behavior alignment reward function to produce policies that are effective according to $r_p$. This design adds robustness against any misspecification of the auxiliary rewards.[3] In the inner-level optimization, by contrast, `Alg` identifies a policy $\theta(\phi, \varphi)$ that is optimal or near-optimal with respect to $r_\phi$ (which combines $r_{\text{aux}}$ through the behavior alignment reward). In the **inner-level optimization**, the algorithm `Alg` works to identify a policy $\theta(\phi, \varphi)$ that is optimal or near-optimal in terms of $r_\phi$ (which incorporates $r_{\text{aux}}$ via the behavior alignment reward). By employing a bi-level optimization structure, several benefits emerge. When $r_{\text{aux}}$ is well-crafted, $r_\phi$ can exploit its detailed information to give `Alg` frequent/dense reward feedback, thus aiding the search for an optimal $\theta^*$. Conversely, if $r_{\text{aux}}$ leads to sub-optimal policies, then the influence of auxiliary rewards can be modulated or decreased accordingly by the optimization process by adjusting $r_\phi$. Consider, for example, a case where the behavior alignment reward function is defined as $r_\phi(s,a) \coloneqq f_{\phi_1}(s,a) + \phi_2 r_p(s,a) + \phi_3 r_{\text{aux}}(s,a)$. In an adversarial setting—where the designer-specified auxiliary reward $r_{\text{aux}}$ may lead to undesired behavior—the bi-level optimization process has the ability to set $\phi_3$ to 0. This effectively allows the behavior alignment reward function $r_\phi$ to exclude $r_{\text{aux}}$ from consideration. Such a bi-level approach to optimizing the parameters of behavior alignment reward functions can act as a safeguard against the emergence of sub-optimal behaviors due to a misaligned auxiliary reward, $r_{\text{aux}}$. This design is particularly valuable because it allows the objective in (2) to leverage the potentially dense reward structure of $r_{\text{aux}}$ to provide frequent action evaluations when the auxiliary reward function is well-specified. At the same time, the approach maintains robustness against possible misalignments.

**Need for Outer Regularization:** The bi-level optimization problem (2) may have multiple optimal solutions for $\phi$—including the trivial solution where $r_{\text{aux}}$ is always ignored. The goal of regularizing the outer-level objective (in the form of the term $\lambda_\gamma \gamma_\varphi$) is to incorporate a prior that adds a preference for solutions, $\phi^*$, that provide useful and frequent evaluative feedback to the underlying RL algorithm. In the next paragraphs, we discuss the need for such a regularizer and motivate its mathematical form. First, recall that *sparse* rewards can pose challenges for policy optimization. An intuitive solution

---

[2]Our framework can be generalized to support state-action dependent discount rates, $\gamma$.

[3]"Misspecification" indicates that an optimal policy for $r_p + r_{\text{aux}}$ may not be optimal for $r_p$ alone.

to this problem could involve biasing the optimization process towards *denser* behavior alignment reward functions, e.g., by penalizing for sparsity of $r_\phi$. Unfortunately, the distinction between sparse and dense rewards alone may not fully capture the nuances of what designers typically consider to be a "good" reward function. This is the case because *a reward function can be dense and still may not be informative*; e.g., a reward function that provides $-1$ to the agent in every non-goal state is dense but fails to provide useful feedback regarding how to reach a goal state. A better characterization of how useful (or informative) a reward function is may be constructed in terms of how *instructive* and *instantaneous* the evaluation or feedback it generates is. We consider a reward function to be *instructive* if it produces rewards that are well-aligned with the designer's goals. A reward function is *instantaneous* if its corresponding rewards are dense, rather than sparse, and are more readily indicative of the optimal action at any given state.[4] Reward functions that are both instructive and instantaneous can alleviate issues associated with settings with sparse rewards and long horizons. To bias our bi-level optimization objective towards this type of reward function, we introduce a regularizer, $\gamma_\varphi$. This regularizer favors solutions that can generate policies with high performance (i.e., high expected return $J$ with respect to $r_p$) *even when the discount factor $\gamma_\varphi$ is small*. To see why, first notice that this regularizer encourages behavior alignment reward functions that provide more instantaneous feedback to the agent. This has to be the case; otherwise, it would be challenging to maximize long-term reward should the optimized alignment reward function be sparse. Second, the regularizer promotes instructive alignment reward functions—i.e., functions that facilitate learning policies that maximize $J$. This is equally crucial: effective policies under the metric $J$ are the ones that align well with the designer's objectives as outlined in the original reward function, $r_p$.

## 4.1 Overcoming Imperfections of Policy Optimization Algorithms

The advantages of the bi-level formulation in (2) extend beyond robustness to sub-optimality from misspecified $r_{\text{aux}}$. Even with a well-specified $r_{\text{aux}}$, RL algorithms often face design choices, such as the bias-variance trade-off, that can induce sub-optimal solutions. Below we present examples to show how *bias* in the underlying RL algorithm may be mitigated by carefully optimizing $r_\phi$ and $\gamma_\varphi$.

**4.1.1 Bias in policy gradients:** Recall that the popular "policy gradient" $\Delta(\theta, r_p)$ is not, in fact, the gradient of any function, and using it in gradient methods may result in biased and sub-optimal policies [44]. However, policy gradient methods based on $\Delta(\theta, r_p)$ remain vastly popular in the RL literature since they tend to be sample efficient [59]. Let $\Delta_\gamma(\theta, r_p)$ denote the *unbiased* policy gradient, where $\Delta_\gamma(\theta, r_p) := \mathbb{E}[\sum_{t=0}^T \gamma^t \psi_\theta(S_t, A_t) \sum_{j=t}^T \gamma^{j-t} r_p(S_j, A_j)]$. We can show that with a sufficiently expressive parameterization, optimized $r_\phi$ and $\gamma_\varphi$ can effectively mimic the updates that would have resulted from using the *unbiased* gradient $\Delta_\gamma(\theta, r_p)$, even if the underlying RL algorithm uses the *biased* "gradient", $\Delta_{\text{on}}(\theta, \phi, \varphi)$, as defined in (3). Detailed proofs are in Appendix A.

**Property 2.** *There exists $r_\phi : \mathcal{S} \times \mathcal{A} \to \mathbb{R}$ and $\gamma_\varphi \in [0, 1)$ such that $\Delta_{\text{on}}(\theta, \phi, \varphi) = \Delta_\gamma(\theta, r_p)$.*

**4.1.2 Off-policy learning without importance sampling:** To increase sample efficiency when evaluating a given policy $\pi_\theta$, it is often useful to use off-policy data collected by a different policy, $\beta$. Under the assumption that $\forall s \in \mathcal{S}, \forall a \in \mathcal{A}, \frac{\pi_\theta(s,a)}{\beta(s,a)} < \infty$, importance ratios $\rho_j := \prod_{k=0}^j \frac{\pi_\theta(s,a)}{\beta(s,a)}$ can be used to adjust the updates and account for the distribution shift between trajectories generated by $\beta$ and $\pi_\theta$. However, to avoid the high variance stemming from $\rho_j$, many methods tend to drop most of the importance ratios and thus only partially correct for the distribution shift—-which can lead to bias [51]. We can show (given a sufficiently expressive parameterization for the behavior alignment reward function) that this type of bias can also be mitigated by carefully optimizing $r_\phi$ and $\gamma_\varphi$.

Let us denote the unbiased off-policy update with full-distribution correction as $\Delta_{\text{off}}(\theta, r_p) := \mathbb{E}_\beta[\sum_{t=0}^T \gamma^t \psi_\theta(S_t, A_t) \sum_{j=t}^T \rho_j \gamma^j r_p(S_j, A_j)]$. Now consider an extreme scenario where off-policy evaluation is attempted *without any correction for distribution shift*. In this situation, and with a slight abuse of notation, we define $\Delta_{\text{off}}(\theta, \phi, \varphi) := \mathbb{E}_\beta[\sum_{t=0}^T \psi_\theta(S_t, A_t) \sum_{j=t}^T \gamma_\varphi^{j-t} r_\phi(S_j, A_j)]$.

**Property 3.** *There exists $r_\phi : \mathcal{S} \times \mathcal{A} \to \mathbb{R}$ and $\gamma_\varphi \in [0, 1)$ such that $\Delta_{\text{off}}(\theta, \phi, \varphi) = \Delta_{\text{off}}(\theta, r_p)$.*

**Remark 1.** *Our method is capable of mitigating various types of algorithmic biases and imperfections in underlying RL algorithms, without requiring any specialized learning rules. Additionally, thanks*

---

[4]E.g., if $r_\phi \approx q^*$, then its corresponding rewards are instantly indicative of the optimal action at any state.

*to the $\gamma_{\varphi^*}$ regularization, it favors reward functions that lead to faster learning of high-performing policies aligned with the designer's objectives, as outlined in the original reward function $r_p$.*

## 5  `BARFI`: Implicitly Learning Behavior Alignment Rewards

Having introduced our bi-level objective and discussed the benefits of optimizing $r_\phi$ and $\gamma_\varphi$, an important question arises: Although $\theta(\phi, \varphi)$ can be optimized using any policy learning algorithm, how can we efficiently identify the optimal $\phi^*$ and $\varphi^*$ in equation (2)? Given the practical advantages of gradient-based methods, one would naturally consider using them for optimizing $\phi^*$ and $\varphi^*$ as well. However, a key challenge in our setting lies in computing $\mathrm{d}J(\theta(\phi, \varphi))/\mathrm{d}\phi$ and $\mathrm{d}J(\theta(\phi, \varphi))/\mathrm{d}\varphi$. These computations require an analytical characterization of the impact that $r_\phi$ and $\gamma_\varphi$ have on the *entire optimization process* of the inner-level algorithm, `Alg`.

In addressing this challenge, we initially focus on an `Alg` that employs policy gradients for updating $\pi_\theta$. Similar extensions for other update rules can be derived similarly. We start by re-writing the expression for $\mathrm{d}J(\theta(\phi, \varphi))/\mathrm{d}\phi$ using the chain rule:

$$\frac{\mathrm{d}J(\theta(\phi, \varphi))}{\mathrm{d}\phi} = \underbrace{\frac{\mathrm{d}J(\theta(\phi, \varphi))}{\mathrm{d}\theta(\phi, \varphi)}}_{(a)} \underbrace{\frac{\mathrm{d}\theta(\phi, \varphi)}{\mathrm{d}\phi}}_{(b)}, \tag{4}$$

where *(a)* is the policy gradient at $\theta(\phi, \varphi)$, and *(b)* can be computed via implicit bi-level optimization, as discussed below.

**Implicit Bi-Level Optimization:**  We compute (4) by leveraging implicit gradients [14, 34, 19], an approach previously employed, e.g., in few-shot learning [38, 49] and model-based RL algorithms [50]. First, observe that when `Alg` converges to $\theta(\phi, \varphi)$, then it follows that

$$\Delta(\theta(\phi, \varphi), \phi, \varphi) = 0. \tag{5}$$

Let $\partial f$ denote the partial derivative with respect to the immediate arguments of $f$, and $\mathrm{d}f$ be the total derivative as before. That is, if $f(x, g(x)) := xg(x)$, then $\frac{\partial f}{\partial x}(x, g(x)) = g(x)$ and $\frac{\mathrm{d}f}{\mathrm{d}x}(x, g(x)) = g(x) + x\frac{\partial g}{\partial x}(x)$. Therefore, taking the total derivative of (5) with respect to $\phi$ yields

$$\frac{\mathrm{d}\Delta(\theta(\phi, \varphi), \phi, \varphi)}{\mathrm{d}\phi} = \frac{\partial \Delta(\theta(\phi, \varphi), \phi, \varphi)}{\partial \phi} + \frac{\partial \Delta(\theta(\phi, \varphi), \phi, \varphi)}{\partial \theta(\phi, \varphi)} \frac{\partial \theta(\phi, \varphi)}{\partial \phi} = 0. \tag{6}$$

By re-arranging terms in (6), we obtain the term *(b)* in (4). In particular,

$$\frac{\partial \theta(\phi, \varphi)}{\partial \phi} = -\left(\frac{\partial \Delta(\theta(\phi, \varphi), \phi, \varphi)}{\partial \theta(\phi, \varphi)}\right)^{-1} \frac{\partial \Delta(\theta(\phi, \varphi), \phi, \varphi)}{\partial \phi}. \tag{7}$$

Furthermore, by combining (7) and (4) we obtain the desired gradient expression for $\phi$:

$$\frac{\partial J(\theta(\phi, \varphi))}{\partial \phi} = -\frac{\partial J(\theta(\phi, \varphi))}{\partial \theta(\phi, \varphi)} \underbrace{\left(\frac{\partial \Delta(\theta(\phi, \varphi), \phi, \varphi)}{\partial \theta(\phi, \varphi)}\right)^{-1}}_{\mathbf{H}} \underbrace{\frac{\partial \Delta(\theta(\phi, \varphi), \phi, \varphi)}{\partial \phi}}_{\mathbf{A}}. \tag{8}$$

Similarly, a gradient expression for $\varphi$ can be derived; the full derivation is detailed in Appendix E. Using $\theta^*$ as shorthand for $\theta(\phi, \varphi)$, we find that the terms $\mathbf{A}$ and $\mathbf{H}$ can be expressed as

$$\mathbf{A} = \mathbb{E}_{\mathcal{D}}\left[\sum_{t=0}^{T} \psi_{\theta^*}(S_t, A_t)\left(\sum_{j=t}^{T} \gamma_\varphi^{j-t} \frac{\partial r_\phi(S_j, A_j)}{\partial \phi}\right)^\top\right], \quad \mathbf{H} = \mathbb{E}_{\mathcal{D}}\left[\sum_{t=0}^{T} \frac{\partial \psi_{\theta^*}(S_t, A_t)}{\partial \theta^*}\left(\sum_{j=t}^{T} \gamma_\varphi^{j-t} r_\phi(S_j, A_j)\right)\right].$$

When working with the equations above, we assume the inverse of $\mathbf{H}^{-1}$ exists. To mitigate the risk of ill-conditioning, we discuss regularization strategies for `Alg` in Appendix D. Notice that equations (8) and (15) are the key elements needed to calculate the updates to $\phi$ and $\varphi$ in our bi-level optimization's outer loop. However, computing $\mathbf{A}$ and $\mathbf{H}$ directly can be impractical for high-dimensional problems due to the need for outer products and second derivatives. To address this, we employ two strategies: *(1)* We approximate $\mathbf{H}^{-1}$ using the Neumann series [40], and *(2)* we calculate (8) and (15) via Hessian-vector products [47], which are readily available in modern auto-diff libraries [46]. These methods eliminate the need for explicit storage or computation of $\mathbf{H}$ or $\mathbf{A}$.

The mathematical approach outlined above results in an algorithm with linear compute and memory footprint, having $O(d)$ complexity, where $d$ is the number of parameters for both the policy and the reward function. Details can be found in Appendix C. We refer to our method as BARFI, an acronym for $\underline{b}$ehavior $\underline{a}$lignment $\underline{r}$eward $\underline{f}$unction's $\underline{i}$mplicit optimization.[5] BARFI is designed to iteratively solve the bi-level optimization problem defined in (2). With policy regularization, the updates to $r_\phi$ and $\gamma_\varphi$ incrementally modify $\theta(\phi, \varphi)$. This enables us to initialize Alg using the fixed point achieved in the previous inner optimization step, further reducing the time for subsequent inner optimizations.

## 6 Empirical Analyses

Our experiments serve multiple purposes and include detailed ablation studies. First, we demonstrate our bi-level objective's efficacy in discovering behavior alignment reward functions that facilitate learning high-performing policies. We focus especially on its robustness in situations where designers provide poorly specified or misaligned auxiliary rewards that could disrupt the learning process (Section 6.1). Second, we present a detailed analysis of the limitations of potential-based reward shaping, showing how it can lead to suboptimal policies (Section 6.2). We then provide a qualitative illustration of the behavior alignment reward function learned by BARFI (Section 6.3). Finally, we evaluate how well BARFI scales to problems with high-dimensional, continuous action and state spaces (Section 6.4).

In the sections that follow, we examine a range of methods and reward combinations for comparison:

- *Baseline RL methods*: We consider baseline RL methods that employ a naive reward combination strategy: $\tilde{r}_{\text{naive}}(s, a) := r_p(s, a) + r_{\text{aux}}(s, a)$. In this case, the auxiliary reward from the designer is simply added to the original reward without checks for alignment. Both the REINFORCE and Actor-Critic algorithms are used for optimization.

- *Potential-based shaping*: To assess how well potential-based reward shaping performs, we introduce variants of the baseline methods. Specifically, we investigate the effectiveness of the reward function $\tilde{r}_\Phi(s, a, s') := r_p(s, a) + \gamma r_{\text{aux}}(s') - r_{\text{aux}}(s)$.

- BARFI: We use REINFORCE as the underlying RL algorithm when implementing BARFI and define $r_\phi(s, a) := \phi_1(s, a) + \phi_2(s) r_p(s, a) + \phi_3(s) r_{\text{aux}}(s, a)$. Our implementation includes a warm-up period wherein the agent collects data for a fixed number of episodes, using $\tilde{r}_{\text{naive}}$, prior to performing the first updates to $\phi$ and $\varphi$ (See Appendix 5 for the complete algorithm).

We evaluate each algorithm across four distinct environments: GridWorld, MountainCar [58], Cart-Pole [16], and HalfCheetah-v4 [9]. These domains offer increasing levels of complexity and are intended to assess the algorithms' adaptability. Furthermore, we examine their performance under a variety of auxiliary reward functions, ranging from well-aligned to misaligned with respect to the designer's intended objectives.

In our experiments, we investigate different types of auxiliary reward functions for each environment: some are action-dependent, while others are designed to reward actions aligned with either effective or ineffective known policies. These functions, therefore, vary in their potential to either foster rapid learning or inadvertently mislead the agent away from the designer's primary objectives, hindering the efficiency of the learning process. Comprehensive details of each environment and their corresponding auxiliary reward functions can be found in Appendix F.

### 6.1 BARFI's Robustness to Misaligned Auxiliary Reward Functions

In this section, we evaluate the performance of various methods for reward combination, particularly in scenarios where auxiliary reward functions can either be well-aligned with a designer's intended goals or be misaligned or poorly specified, thus inadvertently hindering efficient learning. We introduce two types of auxiliary reward functions for CartPole. First, we used domain knowledge to design an $r_{\text{aux}}$ that provides bonuses when the agent's actions align with a known effective policy in this domain. Second, we designed an adversarial example where the auxiliary reward function rewards actions that are consistent with a particularly poorly performing policy. For MountainCar, we first leveraged knowledge about an *energy pumping policy* (i.e., a well-known effective policy [18]) to

---

[5]"BARFI" commonly refers to a type of south-Asian sweet confectionery, typically pronounced as 'bur-fee'.

Table 1: Summary of the performance of various reward combination methods and types of $r_{\text{aux}}$

| Method for Reward Combination | CartPole | | MountainCar | |
|---|---|---|---|---|
| | Well-aligned $r_{\text{aux}}$ | Misaligned $r_{\text{aux}}$ | Well-aligned $r_{\text{aux}}$ (w.r.t. *energy policy*) | Partially-aligned $r_{\text{aux}}$ (w.r.t. *high velocity policy*) |
| BARFI (*our method*) | $487.2 \pm 9.4$ | $475.5 \pm 15.5$ | $0.99 \pm 0.0$ | $0.90 \pm 0.1$ |
| $\tilde{r}_{\text{naive}}$ (*naive reward combination*) | $498.9 \pm 1.0$ | $9.04 \pm 0.2$ | $0.99 \pm 0.0$ | $0.63 \pm 0.1$ |
| $\tilde{r}_{\Phi}$ (*potential-based shaping*) | $8.98 \pm 0.2$ | $500 \pm 0.0$ | $0.00 \pm 0.0$ | $0.00 \pm 0.0$ |

BARFI's performance compared to two baselines that use: $\tilde{r}_{\text{naive}}$ and $\tilde{r}_{\Phi}$, respectively. CartPole uses an action-dependent $r_{\text{aux}}$ function that either rewards agents when actions align with a known effective policy (*well-aligned $r_{aux}$*) or with a poorly-performing policy (*misaligned $r_{aux}$*). MountainCar uses either an action-dependent function aligned with an energy-pumping policy [18] or a partially-aligned function incentivizing higher velocities. BARFI consistently achieves near-optimal performance across scenarios, even if given poorly specified/misaligned auxiliary rewards. Competitors, by contrast, often induce suboptimal policies. Performances significantly below the optimal are shown in red, above.

craft an auxiliary reward function that provides bonuses for actions in line with such a control strategy. We also experimented with a partially-aligned auxiliary function that rewards high velocity—a factor not particularly indicative of high performance.

Table 1 summarizes the results across different reward functions and combination methods. The results suggest that if auxiliary rewards provide positive feedback when agents' actions align with effective policies, naive combination methods perform well. In such cases, auxiliary rewards effectively "nudge" agents towards emulating expert actions. However, our experimental results also indicate that *all* baseline methods are susceptible to poor performance when auxiliary rewards are not well aligned with the designer's goals. We provide more discussion for potential-based shaping in Section 6.2.

> The key takeaway from the experimental results in Table 1 is that BARFI consistently performs well across various domains and under different types of auxiliary rewards. Specifically, when designer-specified feedback is appropriate and can assist in accelerating learning, BARFI efficiently exploits it to produce high-performing policies. Conversely, if auxiliary rewards are misaligned with the designer's intended goals, BARFI is capable of adapting and effectively dismissing "misleading rewards". This adaptability ensures that high-performing policies can be reliably identified. Other methods, by contrast, succeed only in some of these scenarios. Importantly, the unpredictability of whether a given auxiliary reward function will aid or hinder learning makes such alternative methods less reliable, as they may fail to learn effective policies.

## 6.2 Pitfalls of potential-based reward shaping

We now turn our attention to the (possibly negative) influence of action-dependent auxiliary rewards, particularly when used in combination with potential-based reward shaping. Our results in Table 1 reveal a key limitation: potential-based shaping struggles to learn efficient policies even when auxiliary rewards are well-aligned with effective strategies. This shortcoming is attributable to the action-dependent nature of the auxiliary rewards, which compromises the potential shaping technique's guarantee of policy optimality.

As there is no prescribed way for designing potential shaping when $r_{\text{aux}}$ is action-dependent, we use a direct extension of the original formulation [41] by considering $\tilde{r}_{\Phi}(s, a, s', a') := r_p(s, a) + \gamma r_{\text{aux}}(s', a') - r_{\text{aux}}(s, a)$. Furthermore, we designed an auxiliary reward function, $r_{\text{aux}}$, that is well aligned: it provides positive reward signals of fixed magnitude both for $(s, a)$ and $(s', a')$ whenever the agent's actions coincide with the optimal policy. Notice, however, that if $\gamma < 1$, the resultant value from $\gamma r_{\text{aux}}(s', a') - r_{\text{aux}}(s, a)$ is *negative*. Such a negative component may deter the agent from selecting actions that are otherwise optimal, depending on how $r_{\text{aux}}$ and $r_p$ differ in magnitude. Conversely, potential-based shaping can also occasionally perform well under misaligned rewards. In these cases, the shaping function may yield a *positive* value whenever the agent selects an optimal action, which could induce well-performing behaviors.

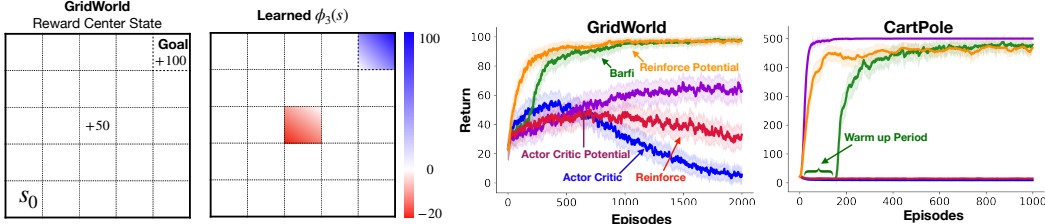

Figure 2: (**Left**) Misaligned $r_{\text{aux}}$ (+50) for entering the center state and $r_p$ (+100) at the goal state. (**Center Left**) Optimized weight $\phi_3(s)$ depicting how the learned reward function penalizes the agent for entering the center state. (**Center Right**) Performances in the GridWorld domain under a misspecified $r_{\text{aux}}$. (**Right**) Performances in the CartPole domain under a $r_{\text{aux}}$ providing positive feedback for mimicking a known ineffective policy.

## 6.3 What does $r_\phi$ learn?

We now investigate BARFI's performance and robustness in the GridWorld when operating under misspecified auxiliary reward functions. Consider the reward function depicted in Figure 2 [left]. This reward function provides the agent with a bonus for visiting the state at the center, akin to providing intermediate feedback to the agent when it makes progress towards the goal. However, such intermediate positive feedback can lead to behaviors where the agent repeatedly cycles around the middle state (i.e., behaviors that are misaligned with the original objective of reaching the goal state at the top right corner of the grid). Importantly, BARFI is capable of autonomously realizing that it should disregard such misleading incentives (Figure 2 [center left]), thereby avoiding poorly-performing behaviors that focus on revisiting irrelevant central states. Similarly, when Cartpole operates under a misspecified $r_{\text{aux}}$ (Figure 2 [right]), BARFI is capable of rapidly adapting (after a warm-up period) and effectively disregarding misleading auxiliary reward signals. These results highlight once again BARFI's robustness when faced with reward misspecification.

## 6.4 Scalability to High-Dimensional Continuous Control

One might wonder whether computing implicit gradients for $\phi$ and $\varphi$ would be feasible in high-dimensional problems, due to the computational cost of inverting Hessians. To address this concern, we leverage Neumann series approximation with Hessian-vector products (See Appendix C) and conduct further experiments, as shown in Figure 3. These experiments focus on evaluating the scalability of BARFI in control problems with high-dimensional state spaces and continuous actions—scenarios that often rely on neural networks for both the policy and critic function approximators. For a more comprehensive evaluation, we also introduced an alternative method named BARFI unrolled. Unlike BARFI, which uses implicit bi-level optimization, BARFI unrolled employs path-wise bi-level optimization. It maintains a complete record of the optimization path to determine updates for $\phi$ and $\varphi$. Further details regarding this alternative method can be found in Appendix C.6.

We conducted experiments on the HalfCheetah-v4 domain and investigated, in particular, a reward function comprising two components with varying weights. This empirical analysis was designed to help us understand how different weight assignments to each reward component could influence the learning process. Specifically, in HalfCheetah-v4, the agent receives a positive reward $r_p$ proportional to how much it moved forward. It also incurs a small negative reward (concretely, an auxiliary reward, $r_{\text{aux}}(s, a) \coloneqq c\|a\|_2^2$, known as a *control cost*) for the torque applied to its joints. A hyperparameter $c$ determines the balance between these rewards. The naive combination of such primary and auxiliary rewards is defined as $\tilde{r}_{\text{naive}}(s, a) = r_p(s, a) + r_{\text{aux}}(s, a)$. Figure 3 [left] shows that the baselines and both variants of BARFI appear to learn effectively. With alternative reward weighting schemes, however, only BARFI and BARFI unrolled show learning progress, as seen in Figure 3 [middle]. It is worth noting that path-wise bi-level optimization can become impractical as the number of update steps in (4) increases, due to growing computational and memory requirements (Figure 3 [right]). Although we do not recommend BARFI unrolled, we include its results for completeness. Additional ablation studies on *(a)* the effect of the inner optimization step; *(b)* Neumann approximations; *(c)* decay of $\gamma$; and *(d)* returns based on $r_\phi$, are provided in Appendix H.

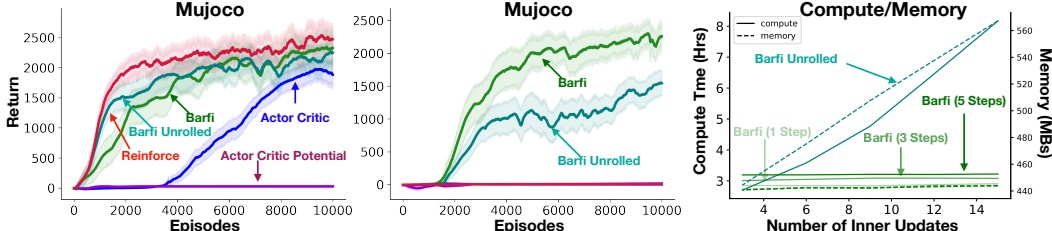

Figure 3: Results for MuJoCo environment. (**Left**) Auxiliary reward is defined to be $-c\|a\|_2^2$, where $c$ is a positive hyperparameter and $a$ is the continuous high-dimensional action vector. (**Middle**) Similar setting as before, but uses an amplified variant of the auxiliary reward: $-4c\|a\|_2^2$. It is worth highlighting that even under alternative reward weighting schemes, both variants of our (behavior-aligned) bi-level optimization methods demonstrate successful learning. Learning curves correspond to mean return over 15 trials, and the shaded regions correspond to one standard error. (**Right**) Required compute and memory for `BARFI unrolled`, compared to `BARFI`, as a function of the number of inner-optimization updates. This figure also showcases `BARFI`'s characteristics under various orders of Neumann approximation.

# 7  Related work

This paper focuses primarily on how to efficiently leverage auxiliary rewards $r_{\texttt{aux}}$. Notice, however, that in the absence of $r_{\texttt{aux}}$, the resulting learned behavior alignment rewards $r_\phi$ may be interpreted as *intrinsic rewards* [70, 71]. Furthermore, several prior works have investigated meta-learning techniques, which are methods akin to the bi-level optimization procedures used in our work. Such prior works have employed meta-learning in various settings, including automatically inferring the effective return of trajectories [68, 62, 7, 71], parameters of potential functions [72, 28, 17], targets for TD learning [69], rewards for planning [54, 23], and even fully specified reinforcement learning update rules [33, 45]. Additionally, various other relevant considerations to effectively learning rewards online have been discussed by Armstrong et al. [5]. Our work complements these efforts by focusing on the reward alignment problem, specifically in settings where auxiliary information is available. An extended discussion on related works can be found in Appendix B. It is worth mentioning that among the above-mentioned techniques, most rely on path-wise meta-gradients. As discussed in Section 6.4, this approach can be disadvantageous as it often performs only one or a few inner-optimization steps, which limits its ability to fully characterize the result of the inner optimization [67]. Further, it requires caching intermediate steps, which increases computational and memory costs. `BARFI`, by contrast, exploits implicit gradients to alleviate these issues by directly characterizing the fixed point of `Alg` induced by learned behavior alignment rewards.

Finally, it is also important to highlight that a concurrent work on reward alignment using bi-level optimization was made publicly available after our manuscript was submitted for peer-reviewing at NeurIPS [10]. While our work analyses drawbacks of potential-based shaping and establishes different forms of correction that can be performed via bi-level optimization, this concurrent work provides complementary analyses on the convergence rates of bi-level optimization, as well as a discussion on its potential applications to Reinforcement Learning from Human Feedback (RLHF).

# 8  Conclusion and Future Work

In this paper, we introduced `BARFI`, a novel framework that empowers RL practitioners—who may not be experts in the field—to incorporate domain knowledge through heuristic auxiliary reward functions. Our framework allows for more expressive reward functions to be learned while ensuring they remain aligned with a designer's original intentions. `BARFI` can also identify reward functions that foster faster learning while mitigating various limitations and biases in underlying RL algorithms. We empirically show that `BARFI` is effective in training agents in sparse-reward scenarios where (possibly poorly-specified) auxiliary reward information is available. If the provided auxiliary rewards are determined to be misaligned with the designer's intended goals, `BARFI` autonomously adapts and effectively disincentivizes their use as needed. This adaptability results in a reliable pathway to identifying high-performing policies. The conceptual insights offered by this work provide RL practitioners with a structured way to design more robust and easy-to-optimize reward functions. We believe this will contribute to making RL more accessible to a broader audience.

## Acknowledgement and Funding Disclosures

We thank Andy Barto for invaluable discussions and insightful feedback on an earlier version of this manuscript, which significantly improved the quality of our work.

This work is partially supported by the National Science Foundation under grant no. CCF-2018372 and by a gift from the Berkeley Existential Risk Initiative.

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

# Behavior Alignment via Reward Function Optimization
## (Supplemental Material)

Table 2: Notations

| Symbol | Description |
|---|---|
| $\theta$ | Parameters for policy $\pi$ |
| $\phi$ | Parameters for reward function |
| $\varphi$ | Parameters for learned $\gamma$ |
| $\pi_\theta, r_\phi, \gamma_\varphi$ | Functional form of policy, reward and $\gamma$ with their respective parameters |
| $\alpha_\theta, \alpha_\phi, \alpha_\varphi$ | Step sizes for the respective parameters |
| $\lambda_\theta, \lambda_\phi, \lambda_\varphi$ | Regularization for policy, reward and $\gamma$ function |
| $\delta$ | Number of on-policy samples collected between subsequent updates to $\phi, \varphi$ |
| $\eta$ | Neumann Approximator Eigen value scaling factor |
| $n$ | Number of loops used in Neumann Approximation |
| `optim` | Any standard optimizer like Adam, RMSprop, SGD, which takes input as gradients and outputs the appropriate update |
| $E$ | Total Number of episodes to sample from the environment |
| $N_i$ | Number of updates to be performed for updating the $\pi$ by `Alg` |
| $N_0$ | Number of initial updates to be peformed |
| $\tau$ | Sample of a trajectory from a full episode |

## A    Proofs for Theoretical Results

In this section, we provide proofs for Property 1, Property 2, and Property 3. For the purpose of these proofs, we introduce some additional notation. To have a unified MDP notation for goal-based and time-based tasks, we first consider that in the the time-based task, time is a part of the state such that Markovian dynamics is ensured.

The (un-normalized) discounted and (un-normalized) undiscounted visitation probability is denoted as

$$d_\gamma^\pi(s, a) := \sum_{t=0}^{T} \gamma^t \Pr(S_t = s, A_t = a; \pi), \tag{9}$$

$$\bar{d}^\pi(s, a) := \sum_{t=0}^{T} \Pr(S_t = s, A_t = a; \pi). \tag{10}$$

We can normalize it so that it is a distribution as follows :

$$d^\pi(s, a) := \frac{\bar{d}^\pi(s, a)}{\sum_{s' \in \mathcal{S}, a' \in \mathcal{A}} \bar{d}^\pi(s', a')}.$$

**Property 1.** *The expected update performed by the biased policy gradient update is same when using the primary reward and reward modified with potential-based shaping, i.e., $\Delta(\theta, \tilde{r}) = \Delta(\theta, r_p)$. Further, the variance of the update when using potential-based reward shaping can be higher than the variance of the update performed using the primary reward, i.e., $\mathrm{Var}\left(\hat{\Delta}(\theta, \tilde{r})\right) \geq \mathrm{Var}\left(\hat{\Delta}(\theta, r_p)\right)$.*

*Proof.* **Part 1: Equality of the expected update**

$$\Delta(\theta, \tilde{r}) = \mathbb{E}_{\pi_\theta} \left[ \sum_{t=0}^{T} \psi_\theta(S_t, A_t) \sum_{j=t}^{T} \gamma^{j-t} \tilde{r}(S_j, A_j) \right]$$

$$= \mathbb{E}_{\pi_\theta} \left[ \sum_{t=0}^{T} \psi_\theta(S_t, A_t) \left( \sum_{j=t}^{T} \gamma^{j-t} \left( r_p(S_j, A_j) + \gamma \Phi(S_{j+1}) - \Phi(S_j) \right) \right) \right]$$

$$= \mathbb{E}_{\pi_\theta} \left[ \sum_{t=0}^{T} \psi_\theta(S_t, A_t) \sum_{j=t}^{T} \gamma^{j-t} r_p(S_j, A_j) \right] + \mathbb{E}_{\pi_\theta} \left[ \sum_{t=0}^{T} \psi_\theta(S_t, A_t) \sum_{j=t}^{T} \gamma^{j-t} (\gamma \Phi(S_{j+1}) - \Phi(S_j)) \right]$$

$$= \Delta(\theta, r_p) + \mathbb{E}_{\pi_\theta} \left[ \sum_{t=0}^{T} \psi_\theta(S_t, A_t) \sum_{j=t}^{T} \gamma^{j-t} (\gamma \Phi(S_{j+1}) - \Phi(S_j)) \right]$$

$$\overset{(a)}{=} \Delta(\theta, r_p) + \mathbb{E}_{\pi_\theta} \left[ \sum_{t=0}^{T} \psi_\theta(S_t, A_t)(\gamma^{T-t+1} \Phi(S_{T+1}) - \Phi(S_t)) \right]$$

$$\overset{(b)}{=} \Delta(\theta, r_p) + \mathbb{E}_{\pi_\theta} \left[ \sum_{t=0}^{T} \psi_\theta(S_t, A_t)(\gamma^{T-t+1} c - \Phi(S_t)) \right]$$

$$\overset{(c)}{=} \Delta(\theta, r_p) + \mathbb{E}_{\pi_\theta} \left[ \sum_{t=0}^{T} (\gamma^{T-t+1} c - \Phi(S_t)) \mathbb{E}_{\pi_\theta} \left[ \psi_\theta(S_t, A_t) | S_t \right] \right]$$

$$\overset{(d)}{=} \Delta(\theta, r_p),$$

where (a) holds because on the expansion of future return, intermediate potential values cancel out, (b) holds because $S_{T+1}$ is the terminal state and potential function is defined to be a fixed constant $c$ for any terminal state [41], (c) holds from the law of total expectation, and (d) holds because,

$$\mathbb{E}_{\pi_\theta} \left[ \psi_\theta(S_t, A_t) | S_t \right] = \sum_{a \in \mathcal{A}} \pi_\theta(S_t, a) \frac{\partial \ln \pi_\theta(S_t, a)}{\partial \theta} = \sum_{a \in \mathcal{A}} \frac{\partial \pi_\theta(S_t, a)}{\partial \theta} = \frac{\partial}{\partial \theta} \sum_{a \in \mathcal{A}} \pi_\theta(S_t, a) = 0.$$

In the stochastic setting, i.e., when using sample average estimates instead of the true expectation, $\gamma^{T-t+1} c - \phi(S_t)$ is analogous to a state-dependent baseline for the sum of discounted future primary rewards. It may reduce or increase the variance of $\Delta(\theta, r_p)$, depending on this baseline's co-variance with $\sum_{j=t}^{T} \gamma^{j-t} r_p(S_j, A_j)$.

**Note:** As we encountered the potential at the terminal state to be $c$, as it is a constant, we will use the value of $c = 0$ in accordance with [41].

**Part 2: Variance characterization**

For this result that discusses the possibility of the variance being higher when using potential-based reward shaping, we demonstrate the result using a simple example. We will consider the single-step case wherein an episode lasts for one time step. That is, the agent takes an action $A_0$ at the starting state $S_0$ and then transitions to the terminal state. Hence, the stochastic update $\hat{\Delta}(\theta, \tilde{r})$ can be written as:

$$\hat{\Delta}(\theta, \tilde{r}) = \psi_\theta(S_0, A_0) \tilde{r}(S_0, A_0)$$
$$= \psi_\theta(S_0, A_0)(r_p(S_0, A_0) - \Phi(S_0)),$$

wherein, we assume that $\Phi$ for terminal states is 0 and similarly, $\hat{\Delta}(\theta, r_p) = \psi_\theta(S_0, A_0) r_p(S_0, A_0)$.

For the purpose of this proof we will consider the case wherein we have a scalar $\theta$, i.e., $\theta \in \mathbb{R}$, such that, $\psi_\theta(., .) \in \mathbb{R}$.

Hence, $\mathrm{Var}\left(\hat{\Delta}(\theta, \tilde{r})\right)$ can be written as:

$$
\begin{aligned}
\mathrm{Var}\left(\hat{\Delta}(\theta, \tilde{r})\right) =& \mathbb{E}\left[\hat{\Delta}(\theta, \tilde{r})^2\right] - \mathbb{E}\left[\hat{\Delta}(\theta, \tilde{r})\right]^2 \\
=& \mathbb{E}\left[(\psi_\theta(S_0, A_0)(r_p(S_0, A_0) - \Phi(S_0)))^2\right] - \mathbb{E}[\psi_\theta(S_0, A_0)(r_p(S_0, A_0) - \Phi(S_0))]^2 \\
\overset{(a)}{=}& \mathbb{E}\left[\psi_\theta(S_0, A_0)^2(r_p(S_0, A_0)^2 + \Phi(S_0)^2 - 2\Phi(S_0)r_p(S_0, A_0))\right] - \mathbb{E}[\psi_\theta(S_0, A_0)(r_p(S_0, A_0))]^2 \\
=& \mathbb{E}\left[\psi_\theta(S_0, A_0)^2(\Phi(S_0)^2 - 2\Phi(S_0)r_p(S_0, A_0))\right] + \\
& \underbrace{\mathbb{E}\left[\psi_\theta(S_0, A_0)^2 r_p(S_0, A_0)^2\right] - \mathbb{E}[\psi_\theta(S_0, A_0)(r_p(S_0, A_0))]^2}_{\mathrm{Var}\left(\hat{\Delta}(\theta, r_p)\right)} .
\end{aligned}
$$

Therefore,

$$
\mathrm{Var}\left(\hat{\Delta}(\theta, \tilde{r})\right) - \mathrm{Var}\left(\hat{\Delta}(\theta, r_p)\right) = \mathbb{E}\left[\psi_\theta(S_0, A_0)^2(\Phi(S_0)^2 - 2\Phi(S_0)r_p(S_0, A_0))\right] .
$$

Subsequently, variance of $\hat{\Delta}(\theta, \tilde{r})$ will be higher than that of $\hat{\Delta}(\theta, r_p)$ if $\mathbb{E}\left[\psi_\theta(S_0, A_0)^2\Phi(S_0)^2\right] - 2\mathbb{E}\left[\psi_\theta(S_0, A_0)^2\Phi(S_0)r_p(S_0, A_0)\right] > 0$.

**Example:** Let us look at an example where the above condition can be true. Let us consider an MDP with a single state and a single-step horizon. In that case, we can consider the variance of the update to the policy at the said state, i.e.,

$$
\mathrm{Var}_\pi\left(\hat{\Delta}(\theta, \tilde{r})\right) - \mathrm{Var}_\pi\left(\hat{\Delta}(\theta, r_p)\right) = \Phi(s)^2\mathbb{E}_\pi\left[\psi_\theta(s, A)^2\right] - 2\Phi(s)\mathbb{E}_\pi\left[\psi_\theta(s, A)^2(r_p(s, A))\right] ,
$$

where $s$ is the fixed state. Hence, the variance of the potential-based method might be more than the variance from using only the primary reward when

$$
\begin{aligned}
\Phi(s)^2\mathbb{E}_\pi\left[\psi_\theta(s, A)^2\right] - 2\Phi(s)\mathbb{E}_\pi\left[\psi_\theta(s, A)^2(r_p(s, A))\right] &> 0 \\
\Phi(s)^2\mathbb{E}_\pi\left[\psi_\theta(s, A)^2\right] &> 2\Phi(s)\mathbb{E}_\pi\left[\psi_\theta(s, A)^2(r_p(s, A))\right] .
\end{aligned}
$$

Further, let us consider the case where $\Phi(s) \neq 0$, because otherwise the variance of the update for those states would be same, and $\Phi(s) > 0$.

$$
\begin{aligned}
\Phi(s)^2\mathbb{E}_\pi\left[\psi_\theta(s, A)^2\right] &> 2\Phi(s)\mathbb{E}_\pi\left[\psi_\theta(s, A)^2(r_p(s, A))\right] \\
\Phi(s)\mathbb{E}_\pi\left[\psi_\theta(s, A)^2\right] &> 2\mathbb{E}_\pi\left[\psi_\theta(s, A)^2(r_p(s, A))\right] .
\end{aligned}
$$

We can see that the above condition can be satisfied by choosing a potential function that might be overly optimistic about the average reward of the state $s$, i.e. any $\Phi(s)$, s.t. $\Phi(s) > 2r_p(s, a) \forall a$ would lead to an increase in variance. A common place where this might be true is the use of an optimal value function (as hinted by [41]) as a baseline for a bad/mediocre policy initially. $\quad\square$

**Property 2.** *There exists $r_\phi : \mathcal{S} \times \mathcal{A} \to \mathbb{R}$ and $\gamma_\varphi \in [0, 1)$ such that $\Delta_{on}(\theta, \phi, \varphi) = \Delta_\gamma(\theta, r_p)$.*

*Proof.* Recall the definition of $\Delta_\gamma(\theta, r_p)$ from Section 4.1:

$$
\Delta_\gamma(\theta, r_p) = \mathbb{E}_{\pi_\theta}\left[\sum_{t=0}^{T}\gamma^t\psi_\theta(S_t, A_t)\sum_{j=t}^{T}\gamma^{j-t}r_p(S_j, A_j)\right] .
$$

Using the law of total expectation,

$$\Delta_\gamma(\theta, r_p) = \mathbb{E}_{\pi_\theta}\left[\sum_{t=0}^{T}\gamma^t\psi_\theta(S_t, A_t)\mathbb{E}_{\pi_\theta}\left[\sum_{j=t}^{T}\gamma^{j-t}r_p(S_j, A_j)\middle| S_t, A_t\right]\right]$$

$$= \mathbb{E}_{\pi_\theta}\left[\sum_{t=0}^{T}\gamma^t\psi_\theta(S_t, A_t)q^{\pi_\theta}(S_t, A_t)\right]$$

$$= \sum_{s\in\mathcal{S}, a\in\mathcal{A}}\sum_{t=0}^{T}\gamma^t\Pr(S_t = s, A_t = a; \pi_\theta)\psi_\theta(s, a)q^{\pi_\theta}(s, a)$$

$$= \sum_{s\in\mathcal{S}, a\in\mathcal{A}}\psi_\theta(s, a)q^{\pi_\theta}(s, a)\sum_{t=0}^{T}\gamma^t\Pr(S_t = s, A_t = a; \pi_\theta)$$

$$= \sum_{s\in\mathcal{S}, a\in\mathcal{A}}\psi_\theta(s, a)q^{\pi_\theta}(s, a)d_\gamma^{\pi_\theta}(s, a). \tag{11}$$

Notice from (9) and (10) that for any $(s, a)$ pair, if $d_\gamma^{\pi_\theta}(s, a) > 0$, then $\bar{d}^{\pi_\theta}(s, a) > 0$ since $\gamma \geq 0$. Therefore, dividing and multiplying by $\bar{d}^{\pi_\theta}(s, a)$ leads to:

$$\Delta_\gamma(\theta, r_p) = \sum_{s\in\mathcal{S}, a\in\mathcal{A}}\bar{d}^{\pi_\theta}(s, a)\psi_\theta(s, a)q^{\pi_\theta}(s, a)\frac{d_\gamma^{\pi_\theta}(s, a)}{\bar{d}^{\pi_\theta}(s, a)}$$

$$= \sum_{s\in\mathcal{S}, a\in\mathcal{A}}\sum_{t=0}^{T}\Pr(S_t = s, A_t = a; \pi_\theta)\psi_\theta(s, a)q^{\pi_\theta}(s, a)\frac{d_\gamma^{\pi_\theta}(s, a)}{\bar{d}^{\pi_\theta}(s, a)}$$

$$= \mathbb{E}_{\pi_\theta}\left[\sum_{t=0}^{T}\psi_\theta(S_t, A_t)q^{\pi_\theta}(S_t, A_t)\frac{d_\gamma^{\pi_\theta}(S_t, A_t)}{\bar{d}^{\pi_\theta}(S_t, A_t)}\right].$$

Now, notice that if $\gamma_\varphi = 0$ and $r_\phi(s, a) = q^{\pi_\theta}(s, a)\frac{d_\gamma^{\pi_\theta}(s, a)}{\bar{d}^{\pi_\theta}(s, a)}$, for all $s \in \mathcal{S}$ and $a \in \mathcal{A}$, then

$$\Delta_{\text{on}}(\theta, \phi, \varphi) = \Delta_\gamma(\theta, r_p).$$

$\square$

**Property 3.** *There exists $r_\phi : \mathcal{S} \times \mathcal{A} \to \mathbb{R}$ and $\gamma_\varphi \in [0, 1)$ such that $\Delta_{\text{off}}(\theta, \phi, \varphi) = \Delta_{\text{off}}(\theta, r_p)$.*

*Proof.* This proof follows a similar technique as the proof for Property 2. Recall the definition of $\Delta_{\text{off}}(\theta, r_p)$:

$$\Delta_{\text{off}}(\theta, r_p) := \mathbb{E}_\beta\left[\sum_{t=0}^{T}\gamma^t\psi_\theta(S_t, A_t)\sum_{j=t}^{T}\gamma^{j-t}\rho_j r_p(S_j, A_j)\right]$$

$$:= \mathbb{E}_\beta\left[\sum_{t=0}^{T}\gamma^t\rho_t\psi_\theta(S_t, A_t)\sum_{j=t}^{T}\gamma^{j-t}\rho_{j-t}r_p(S_j, A_j)\right].$$

Now using the law of total expectations,

$$\Delta_{\text{off}}(\theta, r_p) = \mathbb{E}_\beta \left[ \sum_{t=0}^{T} \gamma^t \rho_t \psi_\theta(S_t, A_t) \mathbb{E}_\beta \left[ \sum_{j=t}^{T} \gamma^{j-t} \rho_{j-t} r_p(S_j, A_j) \middle| S_t, A_t \right] \right]$$

$$= \mathbb{E}_{\pi_\theta} \left[ \sum_{t=0}^{T} \gamma^t \psi_\theta(S_t, A_t) \mathbb{E}_{\pi_\theta} \left[ \sum_{j=t}^{T} \gamma^{j-t} r_p(S_j, A_j) \middle| S_t, A_t \right] \right]$$

$$= \mathbb{E}_{\pi_\theta} \left[ \sum_{t=0}^{T} \gamma^t \psi_\theta(S_t, A_t) q^{\pi_\theta}(S_j, A_j) \right]$$

$$= \sum_{s \in \mathcal{S}, a \in \mathcal{A}} \psi_\theta(s,a) q^{\pi_\theta}(s,a) d_\gamma^{\pi_\theta}(s,a),$$

where the last line follows similar to (11). Now, notice that for any $(s, a)$ pair, the assumption that $\pi_\theta(s,a)/\beta(s,a) < \infty$ for all $s \in \mathcal{S}, a \in \mathcal{A}$, implies $d_\gamma^{\pi_\theta}(s,a)/d_\gamma^\beta(s,a) < \infty$. Further, if $d_\gamma^\beta(s,a) > 0$ it has to be that $d^\beta(s,a) > 0$ as well. Therefore, $d_\gamma^{\pi_\theta}(s,a)/d^\beta(s,a) < \infty$ as well. Multiplying and dividing by $d^\beta(s,a)$ results in:

$$\Delta_{\text{off}}(\theta, r_p) = \sum_{s \in \mathcal{S}, a \in \mathcal{A}} \bar{d}^\beta(s,a) \psi_\theta(s,a) q^{\pi_\theta}(s,a) \frac{d_\gamma^{\pi_\theta}(s,a)}{\bar{d}^\beta(s,a)}$$

$$= \sum_{s \in \mathcal{S}, a \in \mathcal{A}} \sum_{t=0}^{T} \Pr(S_t = s, A_t = a; \beta) \psi_\theta(s,a) q^{\pi_\theta}(s,a) \frac{d_\gamma^{\pi_\theta}(s,a)}{\bar{d}^\beta(s,a)}$$

$$= \mathbb{E}_\beta \left[ \sum_{t=0}^{T} \psi_\theta(S_t, A_t) q^{\pi_\theta}(S_t, A_t) \frac{d_\gamma^{\pi_\theta}(S_t, A_t)}{\bar{d}^\beta(S_t, A_t)} \right].$$

Finally, notice that if $\gamma_\varphi = 0$ and $r_\phi(s,a) = q^{\pi_\theta}(s,a) \frac{d_\gamma^{\pi_\theta}(s,a)}{d^\beta(s,a)}$ for all $s \in \mathcal{S}$ and $a \in \mathcal{A}$,

$$\Delta_{\text{off}}(\theta, \phi, \varphi) = \Delta_{\text{off}}(\theta, r_p).$$

$\square$

**Remark 2.** *Notice that as with any optimization problem, issues of realizability and identifiability of the desired $r_\phi$ must be taken into account. The examples provided in this section aim to highlight the capability of optimized behavior alignment reward functions. In particular, they not only improve and accelerate the learning process but are also capable of inducing updates capable of 'fixing' imperfections in the underlying RL algorithm.*

## B   Extended Related Works

The bi-level objective draws inspiration from the seminal work of Singh et al. [52, 53] that provides an optimal-rewards framework for an agent. Prior works have built upon it to explore search techniques using evolutionary algorithms [43, 21], develop extensions for multi-agent setting [66, 22], and mitigate sub-optimality due to use of inaccurate models [55–57]. Our work also builds upon this direction and focuses on various aspects of leveraging auxiliary rewards $r_{\text{aux}}$, while staying robust against its misspecification.

Apart from specifying auxiliary rewards $r_{\text{aux}}$ directly, other techniques for reward specification include linear temporal logic [35, 11, 65, 39] or reward machines [29–31] that allow exposing the reward functions as a white-box to the agent.

Recent works also explore $\gamma$ that is state-action dependent [63, 48], or establishes connection between $\gamma$ and value function regularization in TD learning Amit et al. [3]. These ideas are complementary to our proposed work and combining these with BARFI remains interesting directions for the future.

The concept of path-based meta-learning was initially popularized for few-shot task learning in supervised learning [15, 42]. Similar path-based approaches have been adopted in reinforcement

learning (RL) in various forms [28, 62, 69, 71]. Initially designed for stochastic gradient descent, these methods have been extended to other optimizers such as Adam [32] and RMSprop [25], treating them as differentiable counterparts [20].

## C  Algorithm

In this section we discuss the algorithm for the proposed method. As the proposed method does behavior alignment reward function's implicit optimization, we name it BARFI. Pseudo-code for BARFI is presented in Algorithm 5. We will first build on some preliminaries to understand the concepts.

### C.1  Vector Jacobian Product

Let us assume that, $x \in \mathbb{R}^d, y \in \mathbb{R}^m, f(x, y) \in \mathbb{R}$. Then, we know that $\partial f(x, y)/\partial x \in \mathbb{R}^d, \partial f(x, y)/\partial y \in \mathbb{R}^m, \partial^2 f(x, y)/\partial y \partial x \in \mathbb{R}^{d \times m}$. Let us also assume that we have a vector $v \in \mathbb{R}^d$, and if we need to calculate the following, we can pull the derivative outside as shown:

$$\underbrace{\underbrace{v}_{\mathbb{R}^d} \underbrace{\frac{\partial^2 f(x, y)}{\partial y \partial x}}_{\mathbb{R}^{d \times m}}}_{\mathbb{R}^m} = \underbrace{\frac{\partial}{\partial y} \underbrace{\left\langle \underbrace{v}_{\mathbb{R}^d}, \underbrace{\frac{\partial f(x, y)}{\partial x}}_{\mathbb{R}^d} \right\rangle}_{\mathbb{R}^1}}_{\mathbb{R}^m}.$$

As we can see, the vector Jacobian product can be broken down into differentiating a vector product but shifting the place of multiplication, in which case we assume that the gradient passes through $v$ w.r.t. $y$ and hence we don't ever have to deal with large multiplications. Also note that the outer partial w.r.t. can easily be handled by autodiff packages. A pseudo-code is show in Algorithm 1.

---

**Algorithm 1:** Jacobian Vector Product

---

1 **Input:** $f(x, y) \in \mathbb{R}^1, x \in \mathbb{R}^d, y \in \mathbb{R}^m, v \in \mathbb{R}^d$
2 $f' \leftarrow \texttt{grad}(f(x, y), x)$
3 $\texttt{jvp} \leftarrow \texttt{grad}(f', y, \texttt{grad\_outputs} = v)$
4 **Return:** jvp

---

### C.2  Neumann Series Approximation for Hessian Inverse

Recall, that for a given real number $\beta \in \mathbb{R}$, such that $0 \leq \beta < 1$, we know that the geomertric series of this has a closed form solution, i.e.,

$$s = 1 + \beta^1 + \beta^2 + \beta^3 + \cdots +$$
$$= \frac{1}{1 - \beta}.$$

Similarly, given we have a value $\alpha$ such that $\beta = 1 - \alpha$, we can write $\alpha^{-1}$ as follows:

$$\frac{1}{1 - \beta} = 1 + \beta + \beta^2 + \beta^3 + \cdots +$$
$$\frac{1}{1 - (1 - \alpha)} = 1 + (1 - \alpha) + (1 - \alpha)^2 + (1 - \alpha)^3 + \cdots +$$
$$\alpha^{-1} = 1 + (1 - \alpha) + (1 - \alpha)^2 + (1 - \alpha)^3 + \cdots +$$
$$\alpha^{-1} = \sum_{i=0}^{\infty} (1 - \alpha)^i.$$

The same can be generalized for a matrix, i.e., given a matrix $\mathbf{A} \in \mathbb{R}^{d \times d}$, we can write $\mathbf{A}^{-1}$ as follows:

$$\mathbf{A}^{-1} = \sum_{i=0}^{\infty} (\mathbf{I} - \mathbf{A})^i.$$

Note for the above to hold, matrix $\mathbf{A}$, where we represent $\texttt{eig}(\mathbf{A})$ as the eigenvalues of matrix $\mathbf{A}$, we should have the following condition to hold, $0 < \texttt{eig}(\mathbf{A}) < 1$. Note here we would regularize $\mathbf{A}$ to ensure that all eigenvalues are positive, and then we can always scale the matrix $\mathbf{A}$, by its biggest eigenvalue to ensure that the above condition holds. Let say $\eta = 1/\max \texttt{eig}(\mathbf{A})$. Then we can write the following:

$$
\begin{aligned}
\mathbf{A}^{-1} &= \frac{\eta}{\eta}\mathbf{A}^{-1} \\
&= \eta(\eta\mathbf{A})^{-1} \\
&= \eta\sum_{i=0}^{\infty}(\mathbf{I} - \eta\mathbf{A})^i.
\end{aligned}
$$

As $\eta\mathbf{A}$ would always satisfy the above condition.

### C.3  Neumann Approximation for Hessian Vector Product

Given we have seen how we can approximate the Inverse of a matrix without relying $O(d^3)$ operations, through Neumann approximation, lets look what needs to be done for our updates. Recall that the update $\phi, \varphi$ (8) and (15) were,

$$
\frac{\partial J(\theta(\phi,\varphi))}{\partial\phi} = -\underbrace{\frac{\partial J(\theta(\phi,\varphi))}{\partial\theta(\phi,\varphi)}}_{v}\underbrace{\left(\frac{\partial\Delta(\theta(\phi,\varphi),\phi,\varphi)}{\partial\theta(\phi,\varphi)}\right)^{-1}}_{\mathbf{H}}\underbrace{\frac{\partial\Delta(\theta(\phi,\varphi),\phi,\varphi)}{\partial\phi}}_{\mathbf{A}}
$$

and

$$
\frac{\partial\left(J(\theta(\phi,\varphi)) - \frac{1}{2}\|\gamma_\varphi\|^2\right)}{\partial\varphi} = -\underbrace{\frac{\partial J(\theta(\phi,\varphi))}{\partial\theta(\phi,\varphi)}}_{v}\underbrace{\left(\frac{\partial\Delta(\theta(\phi,\varphi),\phi,\varphi)}{\partial\theta(\phi,\varphi)}\right)^{-1}}_{\mathbf{H}}\underbrace{\frac{\partial\Delta(\theta(\phi,\varphi),\phi,\varphi)}{\partial\varphi}}_{\mathbf{B}} - \frac{\partial\gamma_\varphi}{\partial\varphi}.
$$

Let us look closely at the update for $\phi$ and we can generalize the updates easily for the case of $\varphi$.

$$
\frac{\partial J(\theta(\phi,\varphi))}{\partial\phi} = -v\mathbf{H}^{-1}\mathbf{A}
$$

We first look at how we can approximate the value of $v\mathbf{H}^{-1}$ efficiently, as we can always make use of the Jacobian Vector product later to get $(v\mathbf{H}^{-1})\mathbf{A}$, as $v\mathbf{H}^{-1}$ becomes a vector. Let us assume we wish to run the Neumann approximation up to $n$ steps, i.e., we want to approximate $\mathbf{H}^{-1}$ up to $n$ order Neumann expansion,

$$
\eta(\eta\mathbf{H}^{-1}) \approx \eta\sum_{i=0}^{n}(I - \eta\mathbf{H})^i \tag{12}
$$

Here we are assuming that the outer optimization for update (1) is for the function $J(\theta(\phi,\varphi))$ and the inner optimization which is represented by the update (3) is $f(\theta(\phi,\varphi),\phi,\varphi)$, i.e.,

$$
\Delta(\theta, r_p) = \frac{\partial J(\theta(\phi,\varphi))}{\partial\theta}, \quad \Delta(\theta,\phi,\varphi) = \frac{\partial f(\theta(\phi,\varphi),\phi,\varphi)}{\partial\theta}.
$$

The most common form in which $f(;B)$ is usually defined is the following:

$$
f(\theta,\phi,\varphi;B) := \frac{1}{|B|}\sum_{\tau\in B}\left[\sum_{t=0}^{T}\log(\pi_\theta(S_t^\tau, A_t^\tau))\sum_{j=t}^{T}\gamma_\varphi^{j-t}r_\phi(S_j^\tau, A_j^\tau)\right].
$$

---

**Algorithm 2:** Vector Hessian Inverse Product for (8) i.e., $v\mathbf{H}^{-1}$

---

1  **Input:**  $\theta, \phi, \varphi, J, f, n, \eta, \mathcal{D}_{\text{off}}, \mathcal{D}_{\text{on}}$
2  $v \leftarrow \texttt{grad}(J(\theta; \mathcal{D}_{\text{on}}), \theta)$
3  $v' \leftarrow \eta \times \texttt{grad}(f(\theta, \phi, \varphi; \mathcal{D}_{\text{off}}), \theta)$
4  **Let:**  $v_0 \leftarrow v, p_0 \leftarrow v$
5  **for**  $i \in [0, n)$ **do**
6    $\big\lvert$  $v_{i+1} \leftarrow v_i - \texttt{grad}(v', \theta, \texttt{grad\_outputs} = v_i)$
7    $\big\lfloor$  $p_{i+1} \leftarrow p_i + v_{i+1}$
8  **Return:** $\eta p_n$ ;          // Approximation of $v\mathbf{H}^{-1}$ as in (12)

---

Similarly this can be defined for $J$, except making use of $r_p$ and problem defined $\gamma$:

$$J(\theta; B) := \frac{1}{|B|} \sum_{\tau \in B} \left[ \sum_{t=0}^{T} \log(\pi_\theta(S_t^\tau, A_t^\tau)) \sum_{j=t}^{T} \gamma^{j-t} r_p(S_j^\tau, A_j^\tau) \right] .$$

Finally, once we have $v\mathbf{H}^{-1}$, we can use the Vector Jacobian Product to calculate $(v\mathbf{H}^{-1})\mathbf{A}$ as described in Algorithm 3:

---

**Algorithm 3:** Update for $\phi$, i.e. (8) i.e., $v\mathbf{H}^{-1}\mathbf{A}$

---

1  **Input:**  $\theta, \phi, \varphi, J, f, n, \eta, \mathcal{D}_{\text{off}}, \mathcal{D}_{\text{on}}$
2  $v \leftarrow \text{Algorithm 2 } (\theta, \phi, \varphi, J, f, n, \eta, \mathcal{D}_{\text{off}}, \mathcal{D}_{\text{on}})$
3  $v' \leftarrow \texttt{grad}(f(\theta, \phi, \varphi; \mathcal{D}_{\text{off}}), \theta)$
4  $\Delta_\phi \leftarrow \texttt{grad}(v', \phi, \texttt{grad\_outputs} = v)$
5  **Return** $\Delta_\phi$

---

We can similarly derive updates for $\varphi$. Note we are not including the different forms of regularizers

---

**Algorithm 4:** Update for $\phi$, i.e. (15) i.e., $v\mathbf{H}^{-1}\mathbf{B}$

---

1  **Input:**  $\theta, \phi, \varphi, J, f, n, \eta, \mathcal{D}_{\text{off}}, \mathcal{D}_{\text{on}}$
2  $v \leftarrow \text{Algorithm 2 } (\theta, \phi, \varphi, J, f, n, \eta, \mathcal{D}_{\text{off}}, \mathcal{D}_{\text{on}})$
3  $v' \leftarrow \texttt{grad}(f(\theta, \phi, \varphi), \mathcal{D}_{\text{off}}), \theta)$
4  $\Delta_\varphi \leftarrow \texttt{grad}(v', \varphi, \texttt{grad\_outputs} = v)$
5  **Return** $\Delta_\varphi$

---

over here to reduce clutter, but adding them is simple.

### C.4  Pseudo Code (Algorithm 5)

Lines 8–10 and 21–23 of Algorithm 5 represent the inner optimization process, and the outer optimization process if from lines 16-17. Lines 8–10 is the initial step of updates to converge to the current values of $\phi, \varphi$, and from there onwards after each update of outer optimization, we consequently update the policy in (21–23). The flow of the algorithm is show in Figure 4.

As discussed in Section D, using regularizers in $\Delta(\theta, \phi, \varphi)$ smoothens the objective $J(\theta(\phi, \varphi))$ with respect to $\phi$ and $\varphi$. This is helpful as gradual changes in $r_\phi$ an $\gamma_\varphi$ can result in gradually changes in the fixed point for the inner optimization. Therefore, for computational efficiency, we initialize the policy parameters from the fixed-point of the previous inner-optimization procedure such that the inner-optimization process may start close to the new fixed-point.

In lines 8–10, the inner optimization for the policy parameters $\theta$ are performed till (approximate) convergence. Note that only trajectories from past interactions are used and no new-trajectories are sampled for the inner optimization.

**Algorithm 5:** `BARFI`: Behavior Alignment Reward Function's Implicit optimization

---

1  **Input:** $J, f, \alpha_\theta, \alpha_\phi, \alpha_\varphi, \eta, n, \delta, \texttt{optim}, E, N_i, N_0,$
2  **Initialize:** $\pi_\theta, r_\phi, \gamma_\varphi$
3  **Initialize:** $\texttt{optim}_\theta \leftarrow \texttt{optim}(\alpha_\theta), \texttt{optim}_\phi \leftarrow \texttt{optim}(\alpha_\phi), \texttt{optim}_\varphi \leftarrow \texttt{optim}(\alpha_\varphi)$
4  $\mathcal{D}_{\text{off}} \leftarrow [\,]$
   # Collect a batch of data for warmup period
5  **for** $e \in [1, N_0)$ **do**
6     |  Generate $\tau_e$ using $\pi_\theta$
7     |  Append $\tau_e$ to $\mathcal{D}_{\text{off}}$
   # Initial training steps using warmup data
8  **for** $i \in [0, N_i + N_0)$ **do**
9     |  Sample a batch of trajectories $B$ from $\mathcal{D}_{\text{off}}$
      # Update policy
10    |  $\theta \leftarrow \theta + \texttt{optim}_\theta(\texttt{grad}(f(\theta, \phi, \varphi; B), \theta))$
   # Start reward alignment
11  **for** $e \in [N_0, E)$ **do**
     # Collect a batch of on-policy data
12    |  $\mathcal{D}_{\text{on}} \leftarrow [\,]$
13    |  **for** $j \in [0, \delta)$ **do**
14    |     |  Generate trajectory $\tau_{e+j}$ using $\pi_\theta$ and append in $\mathcal{D}_{\text{on}}$
15    |  $e \leftarrow e + \delta$
     # Update $r_\phi$ and $\gamma_\varphi$
16    |  $\Delta_\phi \leftarrow$ Algorithm 3$(\theta, \phi, \varphi, J, f, n, \eta, \mathcal{D}_{\text{off}}, \mathcal{D}_{\text{on}})$
17    |  $\Delta_\varphi \leftarrow$ Algorithm 4$(\theta, \phi, \varphi, J, f, n, \eta, \mathcal{D}_{\text{off}}, \mathcal{D}_{\text{on}})$
18    |  $\phi \leftarrow \phi + \texttt{optim}_\phi(\Delta_\phi)$
19    |  $\varphi \leftarrow \varphi + \texttt{optim}_\varphi(\Delta_\varphi)$
20    |  $\mathcal{D}_{\text{off}} \leftarrow \mathcal{D}_{\text{off}} + \mathcal{D}_{\text{on}}$
     # Learn policy for new reward function, initializing from the last
21    |  **for** $i \in [0, N_i)$ **do**
22    |     |  Sample a batch of trajectories $B$ from $\mathcal{D}_{\text{off}}$
         # Update policy
23    |     |  $\theta \leftarrow \theta + \texttt{optim}_\theta(\texttt{grad}(f(\theta, \phi, \varphi; B), \theta))$

---

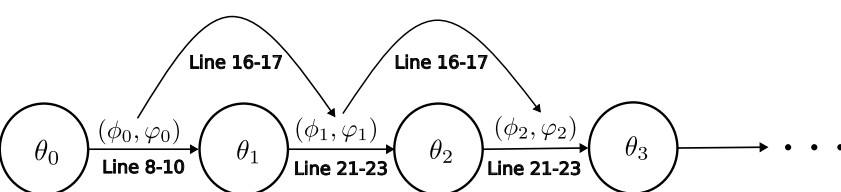

Figure 4: **Algorithm Flow:** The change in different parameters

In Lines 13–14, a new batch $\mathcal{D}_{\text{on}}$ of data is sampled using the policy returned by the inner-optimization process. This data is used to compute $\partial J(\theta(\phi, \varphi))/\partial \theta(\phi, \varphi)$. Existing data $\mathcal{D}_{\text{off}}$ that was used in the inner-optimization process is then used to compute $\partial \theta(\phi, \varphi)/\partial \phi$ and $\partial \theta(\phi, \varphi)/\partial \varphi$. Using these in (8) and (15), the parameters for $r_\phi$ and $\gamma_\varphi$ are updated in Lines 16 and 17, respectively.

Finally, the new data $\mathcal{D}_{\text{on}}$ is merged into the existing data $\mathcal{D}_{\text{off}}$ and the entire process continues.

### C.5  Note on Approximation

An important limitation of the methods discussed above is that $\theta(\phi, \varphi)$ is considered such that $\Delta(\theta(\phi, \varphi), \phi, \varphi) = 0$, i.e., the `Alg` is run to convergence. In practice, we only execute `Alg` for a predetermined number of update steps that need not result in convergence to an optimum *exactly*. However, the impact of this approximation can be bounded by assuming convergence to an $\epsilon$-neighborhood of the optima [49]. Furthermore, due to smoothness in the functional space, slight

changes to $\phi$ and $\varphi$ should result in slight shifts in the optimum $\theta(\phi, \varphi)$. The continuity property allows for improvements in the optimization process: it suffices to initialize the parameters of each inner-loop optimization problem with the final parameters of the approximate fixed point solution, $(\phi, \varphi)$, identified in the previous iteration of the inner loop. The complete resulting algorithm is presented in the appendix as Algorithm 5.

### C.6 Path-wise Bi-level Optimization

An alternative approach for computing the term **(b)** in (4) is possible. The formulation of BARFI described above, based on implicit bi-level optimization, is agnostic to the optimization path taken by Alg. For the sake of completeness, let us also consider a version of BARFI that does take into account the path followed by the inner optimization loop. This is advantageous because it allows us to eliminate the need for the convergence criteria (5). We call this variant BARFI unrolled. The main difference, in this case, is that when computing the term **(b)** in (4), we now consider each inner update step until the point $\theta(\phi, \varphi)$ is reached—where the sequence of steps depends on the specific Alg used for the inner updates. Details are deferred to Appendix C. Notice that this approach results in a path-wise optimization process that can be more demanding in terms of computation and memory. We further discuss this issue, and demonstrate the efficacy of this alternative approach, in the empirical analyses section (Section **??**).

## D Smoothing the objective

To understand why $J(\theta(\phi, \varphi))$ might be ill-conditioned is to note that, often a small perturbation in the reward function doesn't necessarily lead to a change in the corresponding optimal policy. This can lead lack of gradient directions in the neighborhood of $\phi, \varphi$ for gradient methods to be effective. This issue can be addressed by employing common regularization techniques like L2 regularization of the policy parameters or entropy regularization for the policy[6]. We discuss two ways to regularize the objective in the upcoming sections.

### D.1 L2 Regularization

To understand how severely ill-conditioned $J(\theta(\phi, \varphi))$ can be, notice that a small perturbation in the reward function often does not change the corresponding optimal policies or the outcome of a policy optimization algorithm Alg. Therefore, if the parameters of the behavior alignment reward are perturbed from $\phi$ to $\phi'$, it may often be that $J(\theta(\phi, \varphi)) = J(\theta(\phi', \varphi))$ and this limits any gradient based optimization for $\phi$ as $\partial J(\theta(\phi, \varphi))/\partial \phi$ is 0. Similarly, minor perturbations in $\varphi$ may result in no change in $J(\theta(\phi, \varphi))$ either.

Fortunately, there exists a remarkably simple solution: incorporate regularization for the *policy parameters* $\theta$ in objective for Alg in the inner-level optimization. For example, the optimal policy for the following regularized objective $\mathbb{E}_{\pi_\theta}[\sum_{t=0}^{T} \gamma_\varphi^t r_\phi(S_t, A_t)] - \frac{\lambda}{2}\|\theta\|^2$ varies smoothly to trade-off between the regularization value of $\theta$ and the magnitude of the performance characterized by $(r_\phi, \gamma_\varphi)$, which changes with the values of $r_\phi$ and $\gamma_\varphi$. See Figure 5 for an example with L2 regularization.

### D.2 Entropy Regularized

In Section D.1, smoothing of $J(\theta(\phi, \varphi))$ was done by using L2 regularization on the policy parameters $\theta$ in the inner-optimization process. However, alternate regularization methods can also be used. For example, in the following we present an alternate update rule for $\theta$ based on entropy regularization,

$$\Delta(\theta, \phi, \varphi) \coloneqq \mathbb{E}_{\mathcal{D}}\left[\sum_{t=0}^{T} \psi_\theta(S_t, A_t) \sum_{j=t}^{T} \gamma_\varphi^{j-t}\left(r_\phi(S_j, A_j) - \lambda \ln \pi_\theta(S_j, A_j)\right)\right].$$

---

[6]This regularization is performed so as to avoid a noninvertible Hessian as we had discussed in Section 5

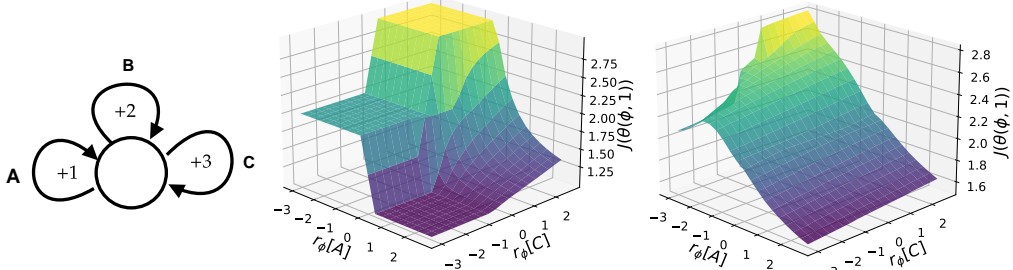

Figure 5: **(Left)** A bandit problem, where the data is collected from a policy $\beta$ that samples action $A$ mostly. **(Middle)** Each point on the 3D surface corresponds to the performance of $\theta(\phi, 1)$ returned by an `Alg` that uses the update rule $\Delta_{\text{off}}(\theta, \phi, 1)$ corresponding to the value of $r_\phi$ for actions $A$ and $C$ in the bottom axes; $r_\theta$ for action $B$ is set to $0$ to avoid another variable in a 3D plot. Notice that small perturbation in $r_\phi$ may lead to no or sudden changes in $J(\theta(\phi, 1))$. **(Right)** Performance of $\theta(\phi, 1)$ returned by an `Alg` that uses the update rule $\Delta_{\text{off}}(\theta, \phi, 1) - \theta$ that incorporates gradient of the L2 regularizer. Vector fields in Figure 1 were also obtained from this setup.

Notice that new update rule for $\phi$ and $\varphi$ can be obtained from steps (4) to (15) with the following $\mathbf{A}$, $\mathbf{B}$, and $\mathbf{H}$ instead, where for shorthand $\theta^* = \theta(\phi, \varphi)$,

$$\mathbf{A} = \mathbb{E}_\mathcal{D} \left[ \sum_{t=0}^{T} \psi_{\theta^*}(S_t, A_t) \left( \sum_{j=t}^{T} \gamma_\varphi^{j-t} \frac{r_\phi(S_j, A_j)}{\partial \phi} \right)^\top \right],$$

$$\mathbf{B} = \mathbb{E}_\mathcal{D} \left[ \sum_{t=0}^{T} \psi_{\theta^*}(S_t, A_t) \left( \sum_{j=t}^{T} \frac{\partial \gamma_\varphi^{j-t}}{\partial \varphi} \left( r_\phi(S_j, A_j) - \lambda \ln \pi_{\theta^*}(S_j, A_j) \right) \right) \right],$$

$$\mathbf{H} = \mathbb{E}_\mathcal{D} \left[ \sum_{t=0}^{T} \frac{\partial \psi_{\theta^*}(S_t, A_t)}{\partial \theta^*} \left( \sum_{j=t}^{T} \gamma_\varphi^{j-t} \left( r_\phi(S_j, A_j) - \lambda \ln \pi_{\theta^*}(S_j, A_j) \right) \right) - \lambda \psi_{\theta^*}(S_t, A_t) \left( \sum_{j=t}^{T} \gamma_\varphi^{j-t} \psi_{\theta^*}(S_j, A_j)^\top \right) \right].$$

## E  Meta Learning via Implicit Gradient: Derivation

The general technique of implicit gradients [14, 34, 19] has been used in a vast range of applications, ranging from energy models [13, 36], differentiating through black-box solvers [61], few-shot learning [38, 49], model-based RL [50], differentiable convex optimization neural-networks layers [4, 2], to hyper-parameter optimization [37, 8, 12, 40]. In this work, we show how implicit gradients can also be useful to efficiently leverage auxiliary rewards $r_a$ and overcome various sub-optimalities.

Taking total derivative in (5) with respect to $\phi$,

$$\frac{d\Delta(\theta(\phi, \varphi), \phi, \varphi)}{d\phi} = \frac{\partial \Delta(\theta(\phi, \varphi), \phi, \varphi)}{\partial \phi} + \frac{\partial \Delta(\theta(\phi, \varphi), \phi, \varphi)}{\partial \theta(\phi, \varphi)} \frac{\partial \theta(\phi, \varphi)}{\partial \phi} = 0. \tag{13}$$

Let us try to understand why the above is true, considering the finite difference approach for this derivative,

$$\frac{d\Delta(\theta(\phi, \varphi), \phi, \varphi)}{d\phi} = \lim_{\|d\phi\| \to 0} \frac{\Delta(\theta(\phi + d\phi, \varphi), \phi + d\phi, \varphi) - \Delta(\theta(\phi, \varphi), \phi, \varphi)}{d\phi}$$

$$= \frac{0 - 0}{d\phi} = 0,$$

$\Delta(\theta(\phi + d\phi, \varphi), \phi + d\phi, \varphi) = \Delta(\theta(\phi, \varphi), \phi, \varphi) = 0$, as $\theta(\cdot, \cdot)$ defines convergence to fixed point.

By re-arranging terms in (13) we obtain the term (b) in (4),

$$\frac{\partial \theta(\phi, \varphi)}{\partial \phi} = - \left( \frac{\partial \Delta(\theta(\phi, \varphi), \phi, \varphi)}{\partial \theta(\phi, \varphi)} \right)^{-1} \frac{\partial \Delta(\theta(\phi, \varphi), \phi, \varphi)}{\partial \phi}. \tag{14}$$

By combining (14) with (4) we obtain the desired gradient expression for $\phi$,

$$\frac{\partial J(\theta(\phi, \varphi))}{\partial \phi} = -\frac{\partial J(\theta(\phi, \varphi))}{\partial \theta(\phi, \varphi)} \underbrace{\left( \frac{\partial \Delta(\theta(\phi, \varphi), \phi, \varphi)}{\partial \theta(\phi, \varphi)} \right)^{-1}}_{\mathbf{H}} \underbrace{\frac{\partial \Delta(\theta(\phi, \varphi), \phi, \varphi)}{\partial \phi}}_{\mathbf{A}},$$

and following similar steps, it can be observed that the gradient expression for $\varphi$,

$$\frac{\partial \big( J(\theta(\phi, \varphi)) - \frac{1}{2}\|\gamma_\varphi\|^2 \big)}{\partial \varphi} = -\frac{\partial J(\theta(\phi, \varphi))}{\partial \theta(\phi, \varphi)} \underbrace{\left( \frac{\partial \Delta(\theta(\phi, \varphi), \phi, \varphi)}{\partial \theta(\phi, \varphi)} \right)^{-1}}_{\mathbf{H}} \underbrace{\frac{\partial \Delta(\theta(\phi, \varphi), \phi, \varphi)}{\partial \varphi}}_{\mathbf{B}} - \frac{\partial \gamma_\varphi}{\partial \varphi},$$

where using $\theta^*$ as a shorthand for $\theta(\phi, \varphi)$ the terms $\mathbf{A}, \mathbf{B}$ and $\mathbf{H}$ can be expressed as

$$\mathbf{A} = \mathbb{E}_\mathcal{D} \left[ \sum_{t=0}^{T} \psi_{\theta^*}(S_t, A_t) \left( \sum_{j=t}^{T} \gamma_\varphi^{j-t} \frac{\partial r_\phi(S_j, A_j)}{\partial \phi} \right)^\top \right], \quad \mathbf{B} = \mathbb{E}_\mathcal{D} \left[ \sum_{t=0}^{T} \psi_{\theta^*}(S_t, A_t) \left( \sum_{j=t}^{T} \frac{\partial \gamma_\varphi^{j-t}}{\partial \varphi} r_\phi(S_j, A_j) \right) \right],$$

$$\tag{15}$$

$$\mathbf{H} = \mathbb{E}_\mathcal{D} \left[ \sum_{t=0}^{T} \frac{\partial \psi_{\theta^*}(S_t, A_t)}{\partial \theta^*} \left( \sum_{j=t}^{T} \gamma_\varphi^{j-t} r_\phi(S_j, A_j) \right) \right] - \lambda.$$

These provide the necessary expressions for updating $\phi$ and $\varphi$ in the outer loop. As $\mathbf{A}$ involves an outer product and $\mathbf{H}$ involves second derivatives, computing them *exactly* might not be practical when dealing with high-dimensions. Standard approximation techniques like conjugate-gradients or Neumann series can thus be used to make it more tractable [40]. In our experiments, we made use of the Neumann approximation to the Hessian Inverse vector product ($\mathbf{A}\mathbf{H}^{-1}$), which requires the same magnitude of resources as the baseline policy gradient methods that we build on top off.

**Algorithm:** Being based on implicit gradients, we call our method `BARFI`, shorthand for *behavior alignment reward function's implicit* optimization. Overall, `BARFI` iteratively solves the bi-level optimization specified in (2) by alternating between using (3) till approximate converge of `Alg` to $\theta(\phi, \varphi)$ and then updating $r_\phi$ and $\gamma_\varphi$. Importantly, being based on (3) for sample efficiency, `Alg` leverages only the past samples and does *not* sample any new trajectories for the inner level optimization. Further, due to policy regularization which smoothens the objective as discussed in D, updates in $r_\phi$ and $\gamma_\varphi$ changes the policy resulting from `Alg` gradually. Therefore, for compute efficiency, we start `Alg` from the policy obtained from the previous inner optimization, such that it is in proximity of the new fixed point. This allows `BARFI` to be both sample and compute efficient while solving the bi-level optimization iteratively online. Pseudo-code for `BARFI` and more details on the approximation techniques can be found in Appendix C.

## F  Environment & Reward Details

The first environment is a **GridWorld** (GW), where the start state is in the bottom left corner and a goal state is in the top right corner. The agent receives an $r_p$ of $+100$ on reaching the goal followed by termination of the episode. The second environment is **MountainCar** (MC) [58], wherein we make use of the sparse reward variant, wherein the agent receives a $+1$ reward on reaching on top of the hill and $0$ otherwise. The third environment is **CartPole** (CP) [16]. Finally, to assess the scalability we pick HalfCheetah-`v4` from **Mujoco** (MJ) suite of OpenAI Gym [9].

For each environment, we define two auxiliary reward functions. For GridWorld, we define the functions: $r_{\text{aux,GW}}^1 := -(s - s_{\text{goal}})^2$, which provides the negative L2 squared distance from the goal position, and $r_{\text{aux,GW}}^1 := 50 \times \mathbf{1}_{s \in \mathcal{S}_{\text{Center}}}$, which provides an additional bonus of $+50$ to the agent along the desired path to the goal state (i.e. the center states). In MountainCar the state is composed of two components: the position $x$, and velocity $\mathtt{v}$. The first auxiliary reward function, $r_{\text{aux,MC}}^1(s, a) := |\mathtt{v}|$, encourages a higher absolute velocity of the car, and the second, $r_{\text{aux,MC}}^1(s, a) := \mathbf{1}_{\text{sign}(\mathtt{v}) = a}$, encourages the direction of motion to increase the magnitude of the velocity (also knows as the *energy pumping policy* [18]). For CartPole, we consider a way to reuse knowledge from a hand crafted policy. CartPole can be solved using a Proportional Derivate (PD) controller [6], hence we tune a PD controller, $\text{PD}^* : \mathcal{S} \to \mathcal{A}$, to solve CartPole for the max possible return. We design two auxiliary

reward functions which make use of this PD controller. The first, $r^1_{\text{aux,CP}}(s,a) := 5 \times \mathbf{1}_{\text{PD*}(s)=a} - (1 - \mathbf{1}_{\text{PD*}(s)=a})$, encourages the agent to match the action of the optimal PD controller, and penalizes it for not matching. The second auxiliary reward function, $r^1_{\text{aux,GW}}(s,a) := -r^1_{\text{aux,CP}}(s,a)$, encourages the agent to do the opposite. In the case of Mujoco, the reward function provided by the environment is itself composed of multiple different functions. We explain the same and the respective auxiliary functions for this case later.

We have considered several forms of information encoded as auxiliary rewards for these experiments. We have heuristic-based reward functions (i.e., $r^1_{\text{aux,GW}}, r^1_{\text{aux,GW}}, r^1_{\text{aux,MC}}$). Reward functions that encode a guess of an optimal policy (i.e., $r^1_{\text{aux,MC}}, r^1_{\text{aux,CP}}$) and reward functions that change the optimal policy (i.e., $r^1_{\text{aux,GW}}, r^1_{\text{aux,CP}}$). We also have rewards that only depend on states (i.e., $r^1_{\text{aux,GW}}, r^1_{\text{aux,GW}}, r^1_{\text{aux,MC}}$) as well as ones that depend on both state and actions (i.e., $r^1_{\text{aux,MC}}, r^1_{\text{aux,CP}}, r^1_{\text{aux,CP}}$). Therefore, we can test if `BARFI` can overcome misspecified auxiliary reward functions and does not hurt performance when well-specified.

**Mujoco Environment** In this experiment, we investigate the scalability of `BARFI` in learning control policies for high-dimensional state spaces with continuous action spaces. In HalfCheetah-`v4` the agent's task is to move forward, and it receives a reward based on its forward movement (denoted as $r_p$). Additionally, there is a small cost associated with the magnitude of torque required for action execution (denoted as $r_{\text{aux}}(s,a) := c|a|^2_2$). The weighting between the main reward and the control cost is pre-defined as $c$ for this environment, and we form the reward as $\tilde{r}(s,a) = r_p(s,a) + r_{\text{aux}}(s,a)$. We explore how an arbitrary weighting choice can cause the agent to fail in learning, while `BARFI` is still able to adapt and learn the appropriate weighting, remaining robust to possible misspecification. We consider two different weightings for the control cost: the first weighting, denoted as $r^1_{\text{aux,MJ}}(s,a) := -c|a|2^2$, uses the default setting, while the second weighting, denoted as $r^1_{\text{aux,MJ}}(s,a) := -4c|a|^2_2$, employs a scaled variant of the first weighting. Additionally, we implement the path-wise bi-level optimization variant i.e., `BARFI Unrolled`. In these experiments, we keep the value of $\gamma$ fixed to isolate the agent's capability to adapt and recover from an arbitrary reward weighting. We will also measure what computational and performance tradeoffs we might have to make between using the implicit version i.e., `BARFI` against, the path-wise version i.e., `BARFI Unrolled`.

# G  Details for the Empirical Results

## G.1  Implementation Details

In this section we will briefly describe the implementation details around the different environments that were used.

**GridWorld (GW):** In the case of GridWorld we made use of the Fourier basis (of Order = 3) over the raw coordinates of agent position in the GridWorld. Details about this could be found in the `src/utils/Basis.py` file.

**MountainCar (MC):** For this environment, to reduce the limitation because of the function approximator we used TileCoding [58], which offers a suitable representation for the MountainCar problem. We used 4 Tiles and Tilings of 5.

**CartPole (CP):** For CartPole also make use of Fourier Basis of (Order = 3), with linear function approximator on top of that.

**MuJoco (MJ):** For this we made use of a neural network with 1 hidden layer of 32 nodes and ReLU activation as the function approximator over the raw observations. The output of the policy is continuous actions, hence we used a Gaussian representation, where the policy outputs the mean of the multivariate Gaussian and we used a fixed diagonal standard deviation, fixed to $\sigma = 0.1$.

**General Details:** All the outer returns are evaluated without any discounting, whereas all the inner optimizations were initialized with $\gamma_\varphi = 0.99$. Hence to do this we made $\varphi$ a single bias unit, initialized to 4.6, and passed through a sigmoid (i.e., $\sigma(4.6) = 0.99$).

For GW, CP and MC $r_\phi$ is defined as below

$$r_\phi(s,a) = \phi_1(s) + \phi_2(s)r_p + \phi_3(s)r_a$$

Wherein $\phi_1, \phi_2, \phi_3$ are scalar outputs of a 3-headed function, in this case simply a linear layer over the states inputs.

Whereas in the case of MJ, we have

$$r_\phi(s, a) = \phi_1 + r_p + \phi_3 r_a$$

Wherein $\phi_1$ is initialized to zero and $\phi_3$ is 1.0 act like bias units.

Gradient normalization was used for all the cases where neural nets were involved (i.e., MJ), and also for MJ we modified the Baseline (REINFORCE) update to subtract the running average of the performance as a baseline to get acceptable performance for the baseline method.

## G.2 Hyper-parameter Selection

As different make use of different function approximators the range of hyper-params can vary we talk about all the above over here.

Best-performing Parameters for different methods and environments are listed where

Table 3: Hyper-parameters for GridWorld

| Hyper Parameter | BARFI Value | REINFORCE Value | Actor-Critic Value |
|---|---|---|---|
| $\alpha_\theta$ | $1 \times 10^{-3}$ | $1 \times 10^{-3}$ | $1 \times 10^{-3}$ |
| $\alpha_\phi$ | $5 \times 10^{-3}$ | — | — |
| $\alpha_\varphi$ | $5 \times 10^{-3}$ | — | — |
| optim | RMSprop | RMSprop | RMSprop |
| $\lambda_\theta$ | 0.25 | 0.25 | 0.25 |
| $\lambda_\phi$ | 0.0625 | — | — |
| $\lambda_\varphi$ | 4.0 | — | — |
| Buffer | 1000 | — | — |
| Batch Size | 1 | 1 | 1 |
| $\eta$ | 0.0005 | — | — |
| $\delta$ | 3 | — | — |
| $n$ | 5 | — | — |
| $N_0$ | 150 | — | — |
| $N_i$ | 15 | — | — |

Table 4: Hyper-parameters for MountainCar

| Hyper Parameter | BARFI Value | REINFORCE Value | Actor-Critic Value |
|---|---|---|---|
| $\alpha_\theta$ | 0.015625 | 0.125 | 0.03125 |
| $\alpha_\phi$ | 0.0625 | — | — |
| $\alpha_\varphi$ | 0.0625 | — | — |
| optim | RMSprop | RMSprop | RMSprop |
| $\lambda_\theta$ | 0.0 | 0.0 | 0.25 |
| $\lambda_\phi$ | 0.0 | — | — |
| $\lambda_\varphi$ | 0.25 | — | — |
| Buffer | 50 | — | — |
| Batch Size | 1 | 1 | 1 |
| $\eta$ | 0.001 | — | — |
| $\delta$ | 3 | — | — |
| $n$ | 5 | — | — |
| $N_0$ | 50 | — | — |
| $N_i$ | 15 | — | — |

**Hyperparameter Sweep** : Here we list the details about how we swept the values for different hyper-params. We used PyTorch [46] for all our implementations. We usually used an optimizer between RMSProp or Adam with default parameters as provided in Pytorch. For $\alpha_\theta \in \{5 \times$

Table 5: Hyper-parameters for CartPole

| Hyper Parameter | BARFI Value | REINFORCE Value | Actor-Critic Value |
|---|---|---|---|
| $\alpha_\theta$ | $1 \times 10^{-3}$ | $1 \times 10^{-3}$ | $5 \times 10^{-4}$ |
| $\alpha_\phi$ | $1 \times 10^{-3}$ | — | — |
| $\alpha_\varphi$ | $5 \times 10^{-3}$ | — | — |
| optim | RMSprop | RMSprop | RMSprop |
| $\lambda_\theta$ | 1.0 | 1.0 | 0.0 |
| $\lambda_\phi$ | 0.0 | — | — |
| $\lambda_\varphi$ | 4.0 | — | — |
| Buffer | 10000 | — | — |
| Batch Size | 1 | 1 | 1 |
| $\eta$ | 0.0005 | — | — |
| $\delta$ | 3 | — | — |
| $n$ | 5 | — | — |
| $N_0$ | 150 | — | — |
| $N_i$ | 15 | — | — |

Table 6: Hyper-parameters for MuJoco

| Hyper Parameter | BARFI Value | REINFORCE Value | Actor-Critic Value |
|---|---|---|---|
| $\alpha_\theta$ | $7.5 \times 10^{-5}$ | $5 \times 10^{-4}$ | $2.5 \times 10^{-4}$ |
| $\alpha_\phi$ | $2.5 \times 10^{-3}$ | — | — |
| $\alpha_\varphi$ | 0.0 | — | — |
| optim | Adam | Adam | Adam |
| $\lambda_\phi$ | 0.0625 | — | — |
| $\lambda_\varphi$ | 0.0 | — | — |
| Buffer | 50 | — | — |
| Batch Size | 1 | 1 | 1 |
| $\eta$ | 0.0005 | — | — |
| $\delta$ | 3 | — | — |
| $n$ | 5 | — | — |
| $N_0$ | 30 | — | — |
| $N_i$ | 15 | — | — |

$10^{-3}, 2.5 \times 10^{-3}, 1 \times 10^{-3}, 5 \times 10^{-4}, 2.5 \times 10^{-4}, 1 \times 10^{-4}, 7.5 \times 10^{-5}\}$ , we use similar ranges for $\alpha_\phi, \alpha_\varphi$ (which tend to be larger). For $\lambda_\theta$ and $\lambda_\phi$, we swept from $[0, 0.25, 0.5, 1.0]$ and for $\lambda_\gamma$ we swept from $[0, 0.25, 1.0, 4.0, 16.0]$. We simply list ranges for different values and later we present sensitivity curves showing that these values are usually robust for BARFI across different methods as we can see from the tables above. $\delta \in [1, 3, 5]$, $n \in [1, 3, 5]$, $N_i \in [1, 3, 6, 9, 12, 15]$, $\eta \in [1 \times 10^{-3}, 5 \times 10^{-4}, 1 \times 10^{-4}]$, $N_0 \in [30, 50, 100, 150]$, buffer $\in [25, 50, 100, 1000]$. $\alpha$ for Tilecoding was adopted from [58] and hence similar ranges were swept in that case. Most sweeps were done with around 10 seeds, and later the parameter ranges were reduced and performed with more seeds.

### G.3 Compute

The computer is used for a cluster where the CPU class is Intel Xeon Gold 6240 CPU @2.60GHz. The total compute required for GW was around 3 CPU years[7], CP also required around 3 CPU years, and MC required around 4 CPU years. For MJ we needed around 5-6 CPU years. In total we utilized around 15-16 CPU years, where we needed around 1 GB of memory per thread.

---

[7] 1 CPU year := Compute equal to running a CPU thread for a year.

# H Extra Results & Ablations

## H.1 Experiment on partially misspecified $r_{\texttt{aux}}$

In these set of experiments, we consider the case where auxiliary reward signals could be helpful only in a few—possibly arbitrary—state-action pairs. In general, we anticipate that solutions in this scenario would be such that assigns weightings, allowing the agent to avoid regions where might be misspecified. Meanwhile, the agent would still make use of the places where is well specified and useful.

We consider another $r_{\mathrm{aux}}$ in the GridWorld domain in which the auxiliary reward is misspecified for a subset of states near the starting position. Meanwhile, it is still well-specified for states near the goal (Figure 6 (a)). Figures 6 (b) and (c) illustrate the learned and the weighting on, showcasing the agent's ability to depict the expected behavior described above.

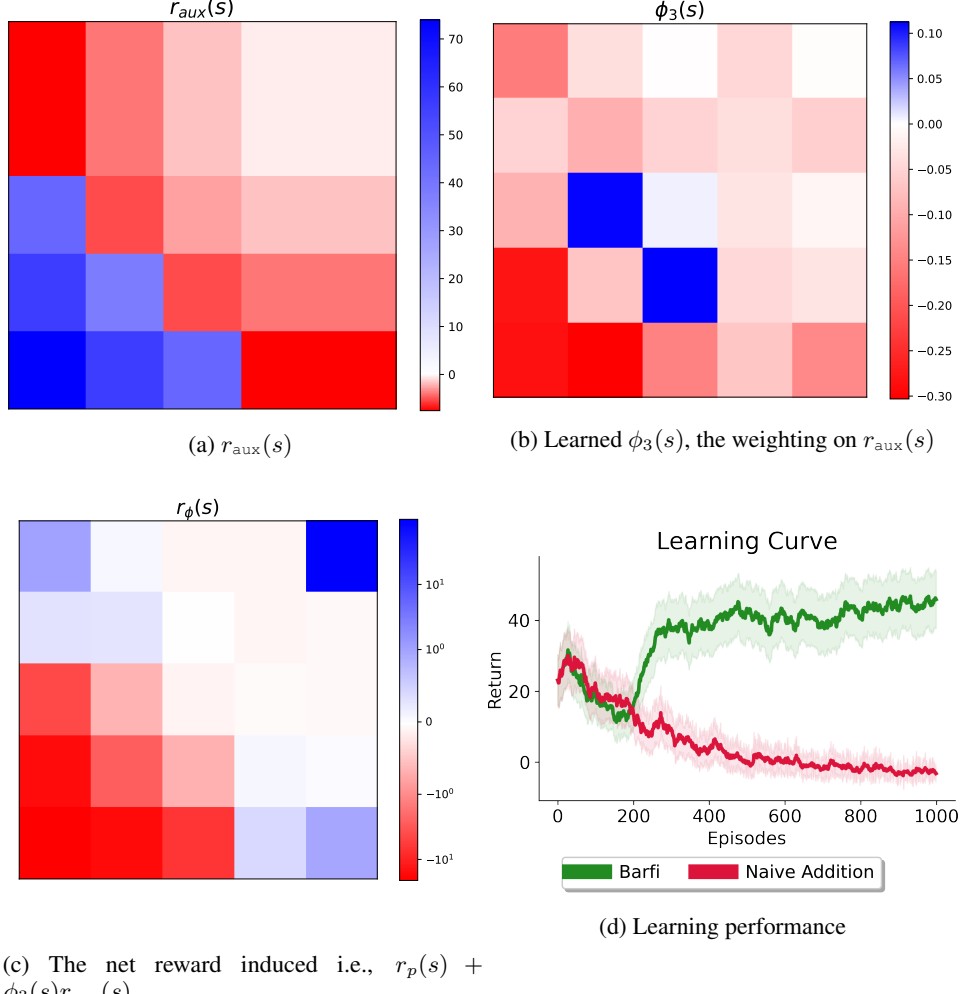

(a) $r_{\mathrm{aux}}(s)$

(b) Learned $\phi_3(s)$, the weighting on $r_{\mathrm{aux}}(s)$

(c) The net reward induced i.e., $r_p(s) + \phi_3(s)r_{\mathrm{aux}}(s)$

(d) Learning performance

Figure 6: 40 random seeds were used to generate the plots. The starting state is at the bottom left and the goal state is at the top right corner. The primary reward $r_p$ is $+100$ when the agent reaches the goal and 0 otherwise. **(a)** A state-dependent $r_{\mathrm{aux}}$ function that is partially misspecified (in the blue region $r_{\mathrm{aux}}$ provides a value equal to the **Manhattan distance**, thereby incentivizing the agent to stay near the start), and partially well specified (in the red region, it is the **negative Manhattan distance** and encourages movement towards the goal). **(b)** The state-dependent weighting $\phi_3(s)$ learned by BARFI negates the positive value from $r_{\mathrm{aux}}$ near the start state. **(c)** The effective reward function $r_p(s) + \phi_3(s)r_{\mathrm{aux}}(s)$ learned by BARFI. **(d)** Learning curves for BARFI, and the baseline that uses a naive addition $(r_p(s) + r_{\mathrm{aux}}(s))$ of the above auxiliary reward.

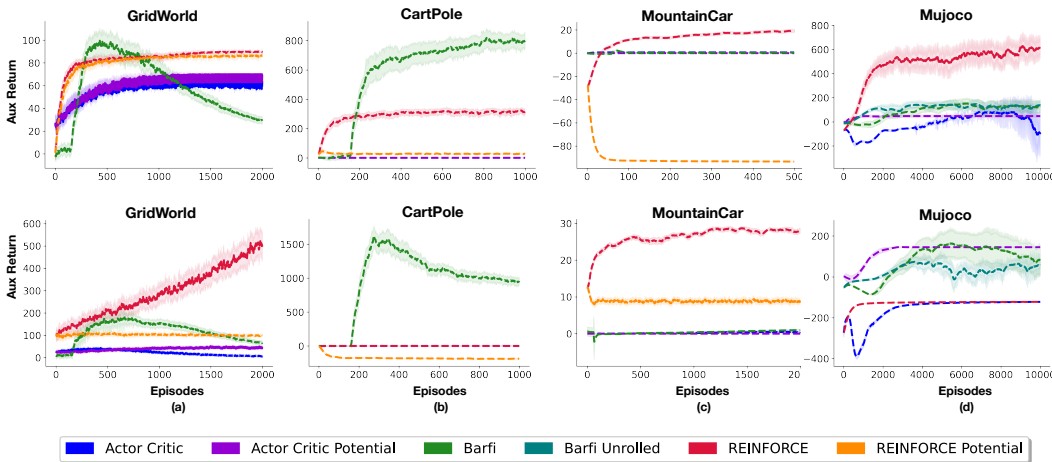

Figure 7: **Return induced by learned reward functions:** This figure illustrates the aux return collected by agent based on the learned $r_\phi$, the curves are chosen based on best-performing curves on $r_p$, and averaged over 20 runs (except 40 for GW). **(a) Top** $- r^1_{\mathtt{aux,GW}}$, **Bottom** $- r^2_{\mathtt{aux,GW}}$, **(b) Top** $- r^1_{\mathtt{aux,CP}}$, **Bottom** $- r^2_{\mathtt{aux,CP}}$, **(c) Top** $- r^2_{\mathtt{aux,MC}}$, **Bottom** $- r^1_{\mathtt{aux,MC}}$, **(d) Top** $- r^1_{\mathtt{aux,MJ}}$, **Bottom** $- r^2_{\mathtt{aux,MJ}}$.

## H.2 Return based on learned $r_\phi$ and $\gamma_\varphi$

Figure 7 and Figure 8 summarize the achievable return based on $r_\phi$ and the $\gamma$ learned by the agent across different domains and reward specification. We observe that REINFORCE often optimizes the naive combination of reward for sure, but that doesn't really lead to a good performance on $r_p$, whereas BARFI does achieve an appropriate return on $r_\phi$, but is also able to successively decay $\gamma$ as the learning progress across different domains. Particularly notice Figure 7 (a) Bottom, where REINFORCE does optimize aux return a lot, but actually fails to solve the problem, as it simply learns to loop around the center state.

## H.3 Ablations

Figure 9 represents the ablation of BARFI on GridWorld with the misspecified reward for its different params. We can see that usually having $\eta = 0.001, 0.0005$, $n = 5$ works for the approximation.

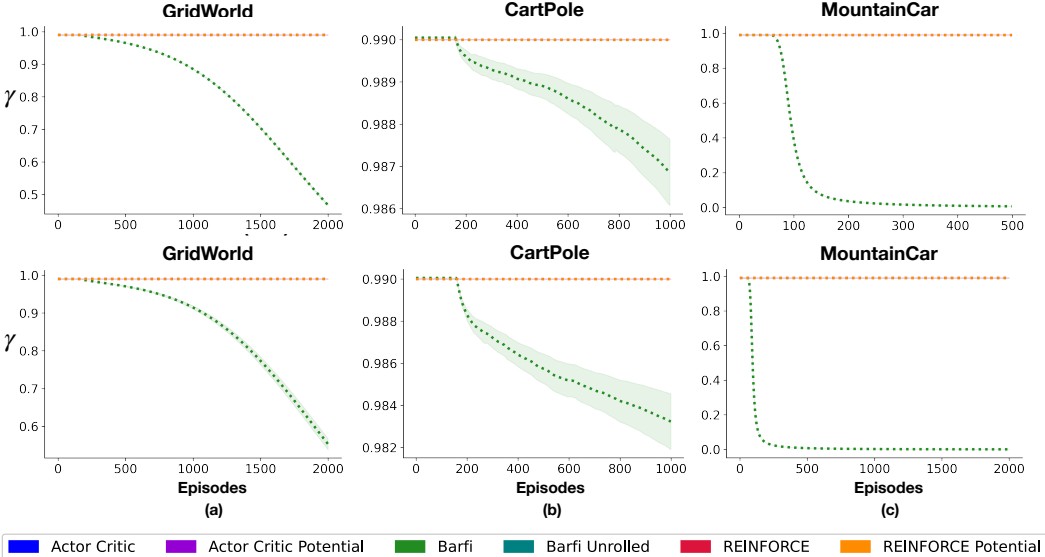

Figure 8: **Learned discounting** $\gamma_\varphi$**:** This figure illustrates the learned $\gamma_\varphi$ for `BARFI` and normal $\gamma$ for other methods, the curves are chosen based on best-performing curves on $r_p$, and averaged over 20 runs (except 40 for GW). **(a) Top** $- r^1_{\mathtt{aux,GW}}$, **Bottom** $- r^2_{\mathtt{aux,GW}}$, **(b) Top** $- r^1_{\mathtt{aux,CP}}$, **Bottom** $- r^2_{\mathtt{aux,CP}}$, **(c) Top** $- r^2_{\mathtt{aux,MC}}$, **Bottom** $- r^1_{\mathtt{aux,MC}}$. Mujoco is not included as the $\gamma$ was not learned in that case. We can observe that the agents start to learn to decay $\gamma$ at the appropriate pace. Note that the curves for methods other than `BARFI` and `BARFI Unrolled` are overlapping as the baselines don't change the value of $\gamma$.

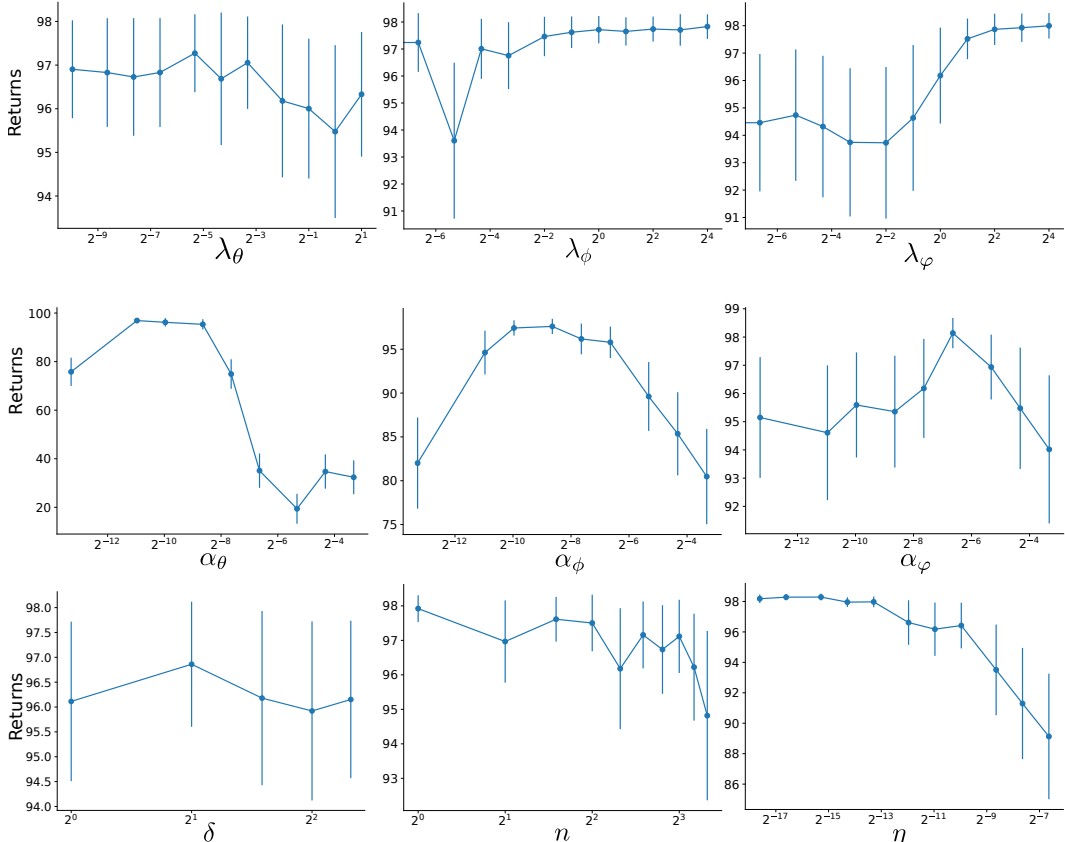

Figure 9: **Sensitivity Curves:** The set of graphs representing the sensitivity of different hyperparams keeping all the other params fixed. The sensitivity is for BARFI in GW with $r^2_{\texttt{aux, GW}}$, i.e., the misspecified reward. We choose the best-performing parameters and vary each parameter to see its influence. The curves are obtained for 50 runs (seeds) in each case, and error bars are standard errors. We can notice that $\alpha_\theta$ and $\alpha_\phi$ can have a large influence, and tend to stay around similar values. $\lambda_{\theta,\phi,\varphi}$ tends to help but doesn't really influence a lot in terms of its magnitude, except larger values of $\lambda_\varphi$ seem to do better. Smaller values of $\eta$ seems to work fine, hence something around $5 \times 10^{-4}, 1 \times 10^{-3}$ usually should suffice. $n, \delta$ can be chosen to around 5 and 3, and usually workout fine. We also defined $N_i = 5 \times \delta$ in this case.

