# OpenReview forum: "Behavior Alignment via Reward Function Optimization"
_NeurIPS.cc/2023/Conference — NeurIPS 2023 spotlight_

### Official Review · Reviewer_tPX3 · 2023-06-22

**Soundness:** 3 good
**Presentation:** 3 good
**Contribution:** 3 good
**Rating:** 6
**Confidence:** 3

**Summary:**

This paper focuses on how to better combine primary rewards that are typically sparse with auxiliary rewards designed by human heuristics to maximize the primary rewards.

The authors propose a bi-level optimization procedure, where the upper level optimizes a behavior alignment reward to guide agent to maximize the primary rewards and meanwhile minimizes the discount value to force the learned rewards to encode long-horizon information, and the lower level can be any policy searching algorithm. They provide both intuitive and theoretical analysis to show that the formulated procedure is not only robust to misspecified auxiliary rewards, but also overcomes the imperfection of policy optimization algorithms, such as biased policy gradients caused by omitted discounted term and no importance sampling in off-policy learning.

Besides the theoretical analysis, the authors also provide two methods to empirically solve this untractable bi-level problem since the optimization of upper level problem requires characterizing the impact of learned rewards and discount value on the entire optimization process. One is a gradient based method that makes use of the implicit gradient denoted as Implicit Bi-level optimization and the other takes into account the complete path followed by the inner optimization loop denoted as Path-wise Bi-level optimization.

The authors also conduct experiments on some case studies to show the robustness and extent to Mujoco tasks to show the scalability to high-dimensional continuous control.

**Strengths:**

1. This paper presents a good motivation to motivate why this work is of great significance and why previous works like potential based methods fail (higher variance and only state-based potential function).

2. This paper dives deep to analyze the advantages of the proposed bi-level objective (Section 3), which is helpful for readers to catch the main idea of this paper.

3. The analysis in Section 3.1 seems novel to me. The proposed bi-level objective can not only address auxiliary reward imperfection but also the imperfection of policy optimization algorithms, which is novel and valuable.

**Weaknesses:**

1. The resulting practical algorithms seem too complex to conveniently apply for higher-dimensional tasks like vision-based tasks.

2. The proposed method introduce a lot of additional hyperparameters as shown in Table 2.3.4.5, which may be hard to tune. For instance, in Table 3, $\alpha_\theta$ is tuned to 0.015625, which does not seem like a common value and may be carefully tuned.

3. The experimental results are not so convincing. In figure 3, Barfi only wins 1 out of 4 tasks. Barfi also cannot outperforms baselines in figure 2. 5.


4. The authors claim that Barfi requires a warm-up stage, after that, Barfi can fastly adapt to find good policies. However, the reasons why Barfi requires warm-up and how does Barfi performs without such warm-up remains unclear.


5. The empirical analysis part is hard to follow, it would be better if the authors could separate some large paragraphs and simplify the abundant detailed task descriptions but focuses on more in-depth analysis.



There is also a recent work that focuses on correcting the imperfect rewards using a bi-level objective in the offline RL setting [1]

[1] Li, J., Hu, X., Xu, H., Liu, J., Zhan, X., Jia, Q. S., & Zhang, Y. Q. Mind the gap: Offline policy optimization for imperfect rewards. ICLR2023.

**Questions:**

Please refer to weaknesses for details.

**Limitations:**

Please refer to weaknesses for details.

---

> ### Author Rebuttal · Authors · 2023-08-10
>
> Thank you for your feedback and for acknowledging the originality and quality of our proposed method. We are glad to hear your positive comments on the practical significance of the problem we address and the novelty of our work. Your positive feedback is encouraging and greatly appreciated.
>
>
> ### High-Dimensional Tasks
> Thank you for bringing this up. BARFI can, indeed, be seamlessly adapted to vision-based tasks. We will emphasize in the paper that our large-scale experiments on MuJoco employed deep neural networks (multi-layer perceptrons) to model both policy and reward functions. We will clarify, thus, that BARFI can be seamlessly adapted to vision-based tasks by simply substituting these MLPs with ConvNets.
>
>
> ### Hyperparameters
> Excellent point. Please notice that the value of $\alpha_\theta$ we reported is simply $2^{-6}$, which can be expressed as $0.015625$ in standard decimal notation. This choice is not arbitrary; rather, it was obtained through a standard hyper-parameter search procedure described in Chapter 3 of Sutton and Barto. This procedure involves exploring candidate step sizes in the form of $2^{-x}$, where $x \in {2, 4, 6, 8}$. Among these options, $2^{-6}$ yielded the best results. We will clarify this process in the main text to ensure readers understand that hyperparameters were fine-tuned using established techniques. Furthermore, we will further emphasize that we also *do* provide sensitivity analysis curves for various hyperparameters, including $\alpha_\theta$ (please see Figure 10 of the appendix).
>
>
> ### Experimental Results
> Thank you for highlighting the need to clarify the specific properties of BARFI investigated in Figures 2 and 3. These figures empirically support our primary claims about BARFI's robustness against potential inaccuracies in auxiliary reward functions, a unique feature of our method described in the Abstract (lines 13--15). In the Introduction (lines 64--66), we further outline our focus on robustness against potential inaccuracies in auxiliary rewards, rather than solely on the goal of accelerating learning.
>
> Figures 2 and 3 effectively showcase our method's success with respect to this goal. For example, in the top left of Figure 3, various methods, including BARFI, perform well with well-defined auxiliary rewards. However, when rewards are misspecified, as in the top right of Figure 3, conventional RL methods (such as REINFORCE and Actor-Critic) often fail, leading to *significantly* suboptimal solutions. BARFI, by contrast, consistently achieves near-optimal performance, thus demonstrating that its performance is not sensitive to whether (or how well) the auxiliary rewards were specified.
>
> When discussing these figures, we will better emphasize their main takeaway message: that predicting the success of an arbitrary RL method under particular (possibly misspecified) auxiliary reward functions is challenging and often infeasible. BARFI remains consistently reliable, regardless of auxiliary reward quality. This robustness is particularly vital in high-stakes applications such as medical uses of RL, where poor performance can lead to significantly risky outcomes.
>
> The experiments in Figures 2 and 3, therefore, serve to illustrate BARFI's robustness and capability of consistently achieving near-optimal performance, regardless of the provided auxiliary reward. This distinguishes it from other techniques, including those based on reward shaping. We will update the paper to better emphasize and discuss these points, as well as how they strongly support BARFI's unique strengths.
>
> ### BARFI Warmup Stage
> We appreciate your observation and will further improve the discussion on BARFI's warmup stage. In Equations 2 and 3, the bi-level objective is defined as the point of convergence with respect to the inner-level objective. When randomly initializing the parameters of the initial policy considered by the algorithm, the warmup phase aids in achieving approximate convergence to the inner-level objective. This is advantageous since subsequent inner-level optimization steps benefit from (a) being able to initialize the policy search process from the previous convergence point; and (b) adapting to gradual changes to the inner-level objective. Empirically, we have observed that this allows BARFI to rapidly identify near-optimal policies in the inner objective. Omitting the warmup phase precludes BARFI from efficiently optimizing the inner-level objective, thereby potentially leading to lower performance. We will update the paper with a more thorough discussion of the properties and requirements of having a warmup phase.
>
> ### Presentation of Empirical Results
> Your feedback on how to better organize the discussion of our empirical results is greatly appreciated. We will follow your suggestion to break down larger paragraphs into shorter ones, and we will move some details about the task descriptions to the footnotes or the appendix.
>
> ### Recent Work
> Thank you for directing us to this potentially relevant paper. It addresses a different problem than ours but also uses similar tools from bi-level optimization. We will expand our Related Work section to discuss this relevant related technique.

---

> ### Comment · Reviewer_tPX3 · 2023-08-13
> **Thanks for the detailed responses**
>
> Thank the authors for the detailed responses. I'm happy to keep my score unchanged (6 weak accept).

---

### Official Review · Reviewer_GcHf · 2023-07-02

**Soundness:** 3 good
**Presentation:** 3 good
**Contribution:** 2 fair
**Rating:** 6
**Confidence:** 3

**Summary:**

This paper proposes a novel framework that employs a bi-level objective for learning a “behavior alignment reward” to find the exact reward specification that elicits the desired behavior in RL. Specifically, the reward function combines auxiliary rewards, defined by a designer’s heuristics, with primary rewards provided by the environment. The experiment shows the performance of the proposed method. However, I have some concerns about this paper. My detailed comments are as follows.

**Strengths:**

1. In order to find the exact reward specification, the authors propose Barfi to dynamically adapts the agent’s policy optimization procedure and address other sub-optimalities introduced by algorithmic design choices.
2. The proposed method allows for the separation of the desired behavior specification from the optimization process with a learned reward function.
3. The authors show how the use of implicit gradients can enhance the efficiency of optimizing the reward function compared to traditional path-wise gradient-based meta optimization methods


**Weaknesses:**

1. Figure 1 explains the effect brought by the proposed method, but does not reflect why the proposed method can change the direction of the gradient and dynamically correct the entire policy optimization process by changing the behavior alignment rewards. It is suggested to visualize the principle of the method in this figure.
2. As for the experiment results, I found that only a few results show that Barfi outperforms other methods, which makes me confused about the effectiveness of the method. Please specify the advantages of the proposed method.
3. The authors compared the proposed method with potential-based reward shaping. Is there any other kind of reward shaping method? It is recommended to compare the proposed method with more relevant work to demonstrate its effectiveness.
4. Some visualized results about the behavior of the agent with the proposed method are required to support the effectiveness.
5. In Figure 9, only two curves in each sub-figure. Please explain where are other curves.
6. This paper provides a detailed theoretical description of the proposed method. However, it is not easy to follow. It would be better to represent the method with a flowchart.



**Questions:**

See the weaknesses.

**Limitations:**

See the weaknesses.

---

> ### Author Rebuttal · Authors · 2023-08-10
>
> We appreciate the reviewer's valuable feedback and acknowledgment of BARFI's general capabilities of dealing with reward misspecification, addressing various potential sub-optimalities introduced by algorithmic design choices, and novel use of implicit gradient techniques to more efficiently optimize reward functions compared to traditional path-wise meta-optimization methods.
>
>
> ### Clarification on Figure 1
> Thank you for your suggestion of an alternative, clearer way of discussing and interpreting Figure 1 to provide further intuition on why and how BARFI changes the direction of the gradients. Extending the paper with this discussion will indeed help better depict BARFI’s dynamic policy optimization procedure. We will update this figure in the final version of the paper by adding arrows indicating how the gradient directions may change over time.
> In terms of additional discussion, we will further clarify that our method can modify the direction of gradients for the inner optimization by adjusting the learned reward function. We hope this explanation, combined with a complementary figure/diagram, helps clarify the mechanism.
>
> ### Experimental Results
> Thank you for highlighting the need to clarify the specific properties of BARFI investigated in Figures 2 and 3. These figures empirically support our primary claims about BARFI's robustness against potential inaccuracies in auxiliary reward functions, a unique feature of our method described in the Abstract (lines 13--15). In the Introduction (lines 64--66), we further outline our focus on robustness against potential inaccuracies in auxiliary rewards, rather than solely on the goal of accelerating learning.
>
> Figures 2 and 3 effectively showcase our method's success with respect to this goal. For example, in the top left of Figure 3, various methods, including BARFI, perform well with well-defined auxiliary rewards. However, when rewards are misspecified, as in the top right of Figure 3, conventional RL methods (such as REINFORCE and Actor-Critic) often fail, leading to *significantly* suboptimal solutions. BARFI, by contrast, consistently achieves near-optimal performance, thus demonstrating that its performance is not sensitive to whether (or how well) the auxiliary rewards were specified.
>
> When discussing these figures, we will better emphasize their main takeaway message: that predicting the success of an arbitrary RL method under particular (possibly misspecified) auxiliary reward functions is challenging and often infeasible. BARFI remains consistently reliable, regardless of auxiliary reward quality. This robustness is particularly vital in high-stakes applications such as medical uses of RL, where poor performance can lead to significantly risky outcomes.
>
> The experiments in Figures 2 and 3, therefore, serve to illustrate BARFI's robustness and capability of consistently achieving near-optimal performance, regardless of the provided auxiliary reward. This distinguishes it from other techniques, including those based on reward shaping. We will update the paper to better emphasize and discuss these points, as well as how they strongly support BARFI's unique strengths.
>
> ### Alternative Reward Shaping Approaches
> The reviewer raises the concern regarding the existence of any alternative well-known and principled way of performing reward shaping, other than via potential-based reward shaping algorithms. In that case, the reviewer argues that we should compare BARFI with them. To the best of our knowledge, however, and after performing a further literature review, no other principled reward shaping methods exist (that are capable of ensuring that the optimal policy will remain unchanged) other than potential-based reward shaping techniques. Kindly let us know if you are aware of any alternative principled techniques of this nature. We will further discuss this point in the paper.
>
> ### Visualization of Results
> Please let us know if we have misunderstood your suggestion, but Figures 1 and 4 were specifically designed to offer insight into the behavior of BARFI's optimization process, as well as the type of reward function that it autonomously identifies to ensure robustness.
>
> Furthermore, to address the reviewer's suggestion of providing further ways of visualizing the solutions produced by BARFI, we have (in response to a comment by reviewer BidQ) performed additional experiments with a novel type of auxiliary reward function: one that is only helpful in a few—possibly arbitrary—state-action pairs, while being misspecified in other. This allows us to directly visualize how the reward functions learned by BARFI adapt to circumvent regions of the state space where auxiliary rewards are misspecified while prioritizing areas where auxiliary rewards are helpful. The results of this experiment are shown in Figures 1(b), (c), and (d), of the attached PDF file.
>
> ### Interpreting Figure 9
> We appreciate your attention to Figure 9. The absence of distinct curves is due to the overlapping of many curves, as all algorithms (except for BARFI) use a constant value of $\gamma$, making them appear indistinguishable. Though it may seem that only two colors are shown, the figure includes colors for all algorithms. We will clarify in the paper that the curves overlap due to the nature of the experiment.
>
> ### Method Flowchart
> We appreciate your feedback regarding augmenting the paper with additional, alternative ways of describing our method. Figure 6 of the appendix already provides a simplified flowchart of how our method works. We will address your suggestion by including more details in Figure 6 (as well as an additional corresponding discussion) in order to produce a more thorough flowchart representation of our algorithm.
>
>
> ### References:
> Zou, H., Ren, T., Yan, D., Su, H., & Zhu, J. (2019). Reward shaping via meta-learning. arXiv preprint arXiv:1901.09330.

---

> > ### Comment · Reviewer_GcHf · 2023-08-16
> > **Thanks for the responses**
> >
> > Main of my concerns are addressed. I am happy to increase the rating to 6 (weak accept).

---

### Official Review · Reviewer_BidQ · 2023-07-07

**Soundness:** 3 good
**Presentation:** 4 excellent
**Contribution:** 3 good
**Rating:** 8
**Confidence:** 4

**Summary:**

The authors propose a method (Barfi) that makes use of auxiliary rewards (provided by the designer), in addition to the rewards already defined for the environment. The method is able to use auxiliary rewards if they are helpful for the task, and avoid the auxiliary signal if they seem to be misspecified. The paper seems well written, for example, it motivates the need for such an algorithm, explains the objective function well, why it's needed, etc. and also seems to have decent experiments.

**Strengths:**



**Weaknesses:**



**Questions:**

On Line 91 -  The work [1] that shows that potential based reward shaping is not suited to Q learning is only for the tabular case, right? Would the results hold for gradient based deep RL algorithms?

It would be interesting to see how Barfi works when the given auxiliary reward is sometimes helpful (for a part of the state-action space) and sometimes misspecified (for another part of the state-action space). Assuming a parameterization like on line 126, would it be possible to show what happens with a Gridworld environment when the auxiliary reward is designed in this way? Does this need a parameterization where $\phi_i$ also depend on $s,a$?

Suggestions related to language and typos:
1. On Line 157 - "due its" -> "due to its"
2. On Line 127 - $\theta_3$ should be $\phi_3$?

References:
1. Potential-based shaping and q-value initialization are equivalent, Wiewiora (2003)

EDIT (16 Aug 2023): updated score from 7->8.


**Limitations:**

---

> ### Author Rebuttal · Authors · 2023-08-10
>
> We appreciate the reviewer's valuable and positive feedback and are encouraged by their appreciation of the quality, relevance, and soundness of our proposed method.
>
> Below, we provide individual responses addressing the reviewer's comments.
>
> ### On using potential-based reward shaping in deep RL settings
> The reviewer is correct: the seminal paper by Ng, Harada, and Russell (1999) discusses the properties of reward shaping only in the tabular setting. Indeed, applying potential-based shaping to problems with function approximation may—as the reviewer suggests—lead to outcomes different than those expected in the tabular case. As an example, the suggested initialization value technique by Wiewiora (2003) is known not to directly translate to the function approximation setting. Having said that, from a practical perspective, the differences between the outcomes of applying reward shaping in the tabular case and in the value function approximate settings appear to be less significant when rich function approximators, like the ones used currently in deep RL. Regarding the properties of shaping in more general settings, we emphasize that while the existing literature on this issue does not provide definite answers about the properties of shaping for the case of *arbitrary* gradient-based deep RL algorithms, in our paper we show  (Property 1) that for policy gradient algorithms such as REINFORCE, the expected update doesn’t change, but can potentially increase the variance. We do appreciate the reviewer's comments and questions regarding a discussion on whether—and to what point—potential-based reward shaping can be applied in deep RL settings. We will update the paper to better discuss these points.
>
>
> ### BARFI's performance under partially misspecified auxiliary reward functions
> This is a great point. The reviewer is correct that the more general setting where auxiliary reward signals could be helpful *only* in a few—possibly arbitrary—state-action pair is indeed an interesting property to investigate.
>
> In general, we anticipate that solutions in this scenario would be such that $\phi_3(s,a)$  assigns weightings, allowing the agent to avoid regions where $r_\texttt{aux}$ might be misspecified. Meanwhile, the agent would still make use of the places where $r_\texttt{aux}$ is well specified and useful.
>
> BARFI's mathematical formulation is expressive enough to facilitate solutions that tackle the challenging setting brought up by the reviewer. Thus, it doesn't require any modification in terms of parameterization and optimization approach to handle this scenario.
>
> To complement this hypothesis and further investigate the setting suggested by the reviewer, we performed additional experiments using a modified grid-world environment. The results are available in the attached PDF. Specifically, we updated the auxiliary reward function in this domain (as depicted in Figure 1(a)) so that $r_\texttt{aux}$ is misspecified for a subset of states near the starting position. Meanwhile, it is still well-specified for states near the goal. Figures 1(b) and (c) illustrate the learned $r_\phi$ and the weighting on $r_\texttt{aux}$, showcasing the agent's ability to depict the expected behavior described above.
>
> ### Feedback on the presentation
> We thank the reviewer for their attention to detail and for the suggestions on how to improve the presentation. We will address them in an updated version of the paper.
>
>
>
>
> ### References
> Ng, A. Y., Harada, D., & Russell, S. (1999). Policy invariance under reward transformations: Theory and application to reward shaping. In ICML (Vol. 99, pp. 278-287).
>
> Wiewiora, E. (2003). Potential-based shaping and Q-value initialization are equivalent. Journal of Artificial Intelligence Research, 19, 205-208.

---

> > ### Comment · Reviewer_BidQ · 2023-08-16
> > **Rebuttal response**
> >
> > I am satisfied with the authors' comments and the experiments. I will increase my score to 8.

---

### Official Review · Reviewer_8Eto · 2023-07-07

**Soundness:** 4 excellent
**Presentation:** 4 excellent
**Contribution:** 4 excellent
**Rating:** 8
**Confidence:** 4

**Summary:**

RL agents often have difficulty learning from sparse rewards, and thus, it is commonplace to introduce an additional auxiliary reward to guide the learning process. However, these heuristic auxiliary rewards can introduce new side effects and biases into learning. In the paper, the authors develop a novel framework for combining the auxiliary reward and true reward in a way that is aligned with the true reward (even given the *learning* process, in contrast to e.g. shaping theorem which says whether the new reward has the same opt policy but does not take into account the potential imperfections in the learning process). In particular, they formulate a bi-level optimization problem in which the outer problem is to learn a combination of the auxiliary and base reward that does well wrt to the true reward given a specific algorithm that solves the inner problem of computing a poilicy given the combined reward.

**Strengths:**

This paper is original, a high-quality, principled solution to a problem with high practical significance. The paper is also very well-written and clear.  Every choice is justified and the authors do a good job of contextualizing their work (e.g. in contrast with potential based reward-shaping).

The experiments are meaningful, well-organized and conveyed. I appreciated that the experiments used a range of realistic auxiliary reward functions including heuristic-based reward functions, reward functions that encode a guess of an optimal policy, ones that change the optimal policy, ones that depend on only states, and ones that depend on both states and actions. The authors report results on robustness, scalability, to high dimensions, and even compute/memory comparisons. In addition, there are several ablation studies in the appendix.

**Weaknesses:**

There is probably more related work to cite in terms of learning from auxiliary rewards, e.g. Inverse Reward Design https://arxiv.org/abs/1711.02827

**Questions:**

n/a

**Limitations:**

The authors address some limitations and extensions in their conclusion.

---

> ### Author Rebuttal · Authors · 2023-08-10
>
> Thank you for your feedback and for acknowledging the originality and quality of our proposed method, including its formal properties, as well as the strengths of our experiments. We are glad you recognized the practical significance of the problem and the novelty of our work. Your positive feedback on the paper are encouraging and greatly appreciated.
>
>
> ### Possibly Relevant Related Work
> Thank you for the suggestion to include further discussion on methods for learning from proxy rewards, such as the paper by Hadfield-Menell et al. (2020). This approach is related and relevant to our goals. It investigates the problem of inferring a designer's "true" underlying objective/reward based on a proxy reward function, an intended decision problem (e.g., an MDP), and a set of possible reward functions. While this is not precisely our objective, their paper sheds light on possible new and improved methodologies for incorporating information into reward functions, as well as possible connections between our work and inverse reward design techniques. In light of your suggestion, we will extend our literature review and add further discussion on methods for reward inference, as well as on reward function identification under various settings, such as when given demonstrations or proxy rewards.
>
>
> ### References
> Hadfield-Menell, D., Milli, S., Abbeel, P., Russell, S., & Dragan, A. (2020). Inverse Reward Design (arXiv:1711.02827). arXiv. https://doi.org/10.48550/arXiv.1711.02827

---

### Author Rebuttal · Authors · 2023-08-10

### Additional figures/experiments (as suggested by reviewer BidQ)

---

### Comment · Area_Chair_tDnC · 2023-08-10
**Author-Reviewer Discussion phase (Aug 10-16)**

Today begins the Author-Reviewer Discussion phase, which lasts 1 week (**Aug 10-16**).

I ask the reviewers to please **carefully read all other reviews and the author responses
and (if appropriate) respond to author responses promptly.**   If you've read the author response, please take the time to leave a comment, even if you have nothing to add.

I also encourage both authors and reviewers to monitor OpenReview for further comments in order to enable as much back-and-forth as possible during this short period.

---

### Decision · Program_Chairs · 2023-09-21

**Decision:**

Accept (spotlight)

**Comment:**

This paper tackles the problem of reward shaping, highlighting issues with the traditional potential-based reward shaping, and proposing an alternative based on bi-level optimization.  Reviewers found the work to be high-quality, the topic interesting and significant, and the approach principled, so I am happy to recommend acceptance.